# Quantifying concentration distributions in redox flow batteries with neutron radiography

Rémy Richard Jacquemond [1,2,10], Maxime van der Heijden [1,10], Emre Burak Boz [1,3,10], Eric Ricardo Carreón Ruiz [4], Katharine Virginia Greco [5,6], Jeffrey Adam Kowalski [5,6], Vanesa Muñoz Perales [7], Fikile Richard Brushett [6], Kitty Nijmeijer [2,8], Pierre Boillat [4,9] & Antoni Forner-Cuenca [1,3,6] ✉

The continued advancement of electrochemical technologies requires an increasingly detailed understanding of the microscopic processes that control their performance, inspiring the development of new multi-modal diagnostic techniques. Here, we introduce a neutron imaging approach to enable the quantification of spatial and temporal variations in species concentrations within an operating redox flow cell. Specifically, we leverage the high attenuation of redox-active organic materials (high hydrogen content) and supporting electrolytes (boron-containing) in solution and perform sub-tractive neutron imaging of active species and supporting electrolyte. To resolve the concentration profiles across the electrodes, we employ an in-plane imaging configuration and correlate the concentration profiles to cell performance with polarization experiments under different operating conditions. Finally, we use time-of-flight neutron imaging to deconvolute concentrations of active species and supporting electrolyte during operation. Using this approach, we evaluate the influence of cell polarity, voltage bias and flow rate on the concentration distribution within the flow cell and correlate these with the macroscopic performance, thus obtaining an unprecedented level of insight into reactive mass transport. Ultimately, this diagnostic technique can be applied to a range of (electro)chemical technologies and may accelerate the development of new materials and reactor designs.

Contemporary energy storage technologies do not fulfil the stringent performance and cost requirements of the current and future electrical grid[1–3]. Developing cost-effective and high-performance storage platforms is paramount to integrate intermittent renewable energy technologies into the energy network[4]. Among the existing battery technologies, redox flow batteries (RFBs) have emerged as a promising candidate for large-scale energy storage. Intrinsic to their design, RFBs offer opportunities for economic scaling as the electrolyte reservoirs and the reactor size can be independently scaled, decoupling the power rating and energy capacity[5]. Furthermore, flow batteries are easy to manufacture, can utilize various electrolyte chemistries, are easily recyclable and generally safer than lithium-ion batteries[5,6]. Common RFB architectures consist of an electrochemical stack with alternating separators and bipolar plates between electrodes where the electrical power is generated, and external storage tanks containing the electrolyte solutions where the energy is stored. The electrolyte is pumped

through the reactor and is composed of dissolved supporting salt and redox active materials that can be reversibly converted between different oxidation states. The redox reactions are sustained at the surface of the porous electrodes through which the electrolyte continuously flows, and an ionic current is carried through the separator to maintain the electroneutrality in each half-cell[7]. However, electrochemical losses (i.e., ohmic, charge transfer and mass transfer overpotentials) impact the overall efficiency and cost of the battery, which challenges their economic competitiveness[8]. Thus, materials research for RFBs is focused on advancing the component properties to improve the overall performance and durability of the system[9–11].

The stack performance and lifetime are governed by the operating conditions of the reactor and properties of the individual cell components and their complex interplay. The physicochemical properties of the electrolyte and the main cell components (porous electrodes, separator), together with the reactor design (flow field geometry, transport lengths) determine the available surface area for reactions and control the transport of mass and charge. The electrode microstructure (and to some extend the reactor design) dictates the trade-off between the pressure drop and the accessible surface area[12], whereas the separator type, chemical composition and nanostructure[13] impact the RFB performance by regulating the transport of species between the positive and negative compartments. Conventionally, the cell performance is assessed using electrochemical diagnostic tools (e.g., polarization curves, charge-discharge capacity and electrochemical impedance spectroscopy) together with ex-situ characterization methods (e.g., scanning electron microscopy with energy dispersive X-ray analysis) where novel materials are macroscopically evaluated by empirical comparison (internal surface area, porosity, tortuosity) with the current state-of-the-art[4]. Although this is a valid approach to screen promising material candidates, valuable information is lost due to the complexity of identifying local performance-limiting factors related to electrode utilization and mass transport within the reactor (i.e., flow fields, porous electrodes and separator). From the perspective of an electrochemical diagnostic technique, the cell is a homogenous system, a so-called "black box", with inputs and outputs in the form of current or voltage. Electrochemical cells, however, are anisotropic systems with an inhomogeneous distribution of reaction rate, current and species concentration within the reactor volume. This gives rise to a myriad of limiting phenomena in the cell that cannot be observed with conventional electrochemical techniques, including gas evolution, improper electrode wetting, local depletion of species concentration, membrane crossover and precipitation. Thus, to properly correlate the material properties to the device performance, the development of locally resolved characterization methods is essential[14–16].

The most straightforward route to access information at a microscopic level is employing *operando* imaging of electrochemical systems in tandem with complementary electrochemical diagnostics, which has been instrumental in the rapid development of polymer electrolyte fuel cells[17,18] and lithium-ion batteries[19,20]. Over the last decade, several groups have developed imaging and spectroscopic approaches that enable *operando* characterization of RFBs. In the following, we describe a set of representative examples, and not an exhaustive list of advances, of *operando* characterization of redox flow cells. Wong et al. applied fluorescence microscopy and particle velocimetry to a quinone-based flow battery to quantitively image active species and velocity distributions near the electrode-flow field interface and highlighted inhomogeneous flow profiles within the porous electrodes[21]. This technique holds promise to image flow velocity profiles and to resolve the electrochemical activity at the electrode-electrolyte interface. However, for this technique to be applied, fluorescent species are necessary, the electrochemical cell must be modified with a transparent window and the technique is restricted to a limited penetration depth through porous carbon or metallic electrodes. Jervis et al., Tariq et al., Eifert et al. and Köble et al. employed X-ray tomographic microscopy to visualize gas pockets within the liquid electrolyte[22–25]. The authors successfully separated the carbon electrode phase from the liquid electrolyte and the gas pockets present in the porous electrodes, showing incomplete wetting of carbon fibre electrodes. X-ray imaging can yield 3D reconstructions of electrode microstructures and can uncover wetting dynamics in porous media. However, despite being considered non-destructive, high X-ray doses are generally needed to obtain high temporal and spatial resolutions, while low doses are necessary to stay under the X-ray-induced degradation threshold of the investigated materials. Thus, a delicate trade-off between resolution and exposure time must be used[26–28]. Zhao et al. coupled in situ nuclear magnetic resonance and electron paramagnetic resonance to track reaction mechanisms occurring within the electrolyte[29,30]. It is an effective technique to monitor the state-of-charge of the electrolyte within the tanks and to track chemical transformations; however, nuclear magnetic resonance-based techniques require challenging cell-design modifications to probe the concentration within the reactor volume. While these techniques have provided important insights into flow distributions, electrode wetting, electrolyte state-of-charge and degradation, a technique that enables *operando*, reactor-level and non-invasive quantification of local concentrations within the electrochemical cell is necessary to understand how the microscopic properties of materials impact the cell performance.

Neutron radiography is an alternative technique to fulfil these requirements and has been successfully applied to a variety of electrochemical devices for more than two decades[31]. Neutron imaging is a noninvasive technique that is suitable to study systems during extended operation. Neutrons are electrically neutral particles and therefore offer a high penetration depth even through high atomic weight elements, as neutrons only interact with the nuclei and not with the electron cloud, requiring no or minimal modification of the electrochemical cell design for imaging[17]. One of the first studies was reported by Mosdale et al. who determined the water content within the membrane of an operando proton exchange membrane fuel cell[32]. Due to the through-plane configuration of their setup, the authors designed a cell with neutron-transparent components, enabling selective investigation of the membrane hydration. Following this work, neutron radiography was mainly applied to develop the understanding of proton exchange membrane fuel cells[17,31,33–36]. The interest of using neutron imaging to study fuel cells stems for the high attenuation of hydrogen atoms in liquid water and low attenuation of gases ($O_2$, air, $H_2$), alongside the high neutron penetration through the materials constituting the cell. Thus, the use of neutron imaging improved the fundamental understanding of water management in fuel cells under various operating conditions[37–40]. Additionally, neutron imaging studies were carried out for a range of other electrochemical technologies such as lithium-ion batteries[19,41–44], electrolyzers[45,46] and alkaline batteries[44,47]. More recently, Clement et al. used neutron radiography to study the gas evolution and correlated it to the properties of the electrode material during the charging process of a vanadium RFB[48]. We hypothesize that neutron imaging can offer more than the contrast between liquid, gas and solid phases, enabling vizualization of the dissolved species within the electrode pore space[49,50]. Unlike X-rays, neutrons have no marked trend in the distribution of mass attenuation coefficients across the periodic table, so contrast between elements does not follow a predictable trajectory. However, several elements (e.g., H, Li, B), which are central to redox flow batteries, feature high attenuation with the neutron beam and make neutrons highly suited to study organic redox molecules[51,52], which are of interest for various electrochemical technologies such as RFBs and $CO_2$-electrolyzers. These aspects make it possible to perform neutron radiography of electrochemical systems and to obtain contrast between the redox active molecules and the supporting salts, opening an avenue to study

in situ or *operando* motion of species in performance-defining materials such as separators and porous electrodes.

Here we explore, for the first time, the use of neutron imaging to reveal reactive transport phenomena and concentration distributions in non-aqueous redox flow batteries (NAqRFBs) with the model redox species (2,2,6,6-tetramethylpiperidin-1-yl)oxyl (TEMPO) to demonstrate this approach. NAqRFBs are an attractive option for energy storage because of their larger electrochemical stability windows and their use of organic molecules made from earth-abundant elements. In this work, neutron radiography is used to extract concentration profiles of redox-active species or supporting salts within the reactor of an *operando* NAqRFB. We perform in-plane transmission neutron imaging of NAqRFBs in two beamlines offering distinct capabilities in terms of resolution and species identification. Compared to conventional through-plane imaging, in-plane imaging can reveal the concentration gradients emerging from the flow field channels towards the separator, provided appropriate spatial resolution is achievable. The NEUTRA beamline operates with thermal neutrons (white beam) allowing the cumulative concentration profiles of the active materials and the supporting salts to be determined with a high spatial resolution over the region of interest. The ICON beamline operates with cold neutrons and utilizes the time-of-flight neutron imaging (ToF-NI) as a spectral technique to image the reactor via energy-resolved neutron radiography, but the use of this technique results in a drop in temporal resolution caused by the use of a pulsed source. ToF-NI enables the deconvolution of concentration profiles of several species in the electrolyte (i.e., active species and supporting ions), which can resolve the movement of species between half cells under a voltage bias, providing insight into the coupled transport phenomena within the reactor area.

In the first part of this work the experiments performed at the NEUTRA beamline are described during RFB polarization (Fig. 1a) to extract concentration profiles of TEMPO in its neutral and oxidized form, dissolved in solution with a low neutron attenuating supporting salt, potassium hexafluorophosphate ($KPF_6$) or with a highly attenuating counter-ion, tetrafluoroborate ($BF_4^-$). To this end, calibration curves are obtained for all species used in this work, after which the cells are imaged during operation and concentration profiles of dissolved species in the reactor volume are extracted. In the second part of this work (Fig. 1b), polarization experiments are performed using TEMPO with $BF_4^-$ as a counter-ion at the ICON beamline. First, the neutron attenuation of the electrolyte solutions are calibrated at different neutron energies prior to *operando* imaging. By utilizing the principle of energy-dependency of their attenuation coefficients, we quantify the concentration change of active species and supporting salt separately during operation and reveal the dominant transport mechanisms within the electrodes and between half cells under voltage stimuli. This approach pushes the limits of neutron imaging for electrochemical systems by probing concentration profiles and species movement in an operating flow cell and we hope it will serve as a guide for researchers intending to perform species-sensitive *operando* neutron imaging.

## Results and discussion

First, we discuss the results of the white-beam imaging obtained at the NEUTRA beamline, followed by the ToF-NI performed at the ICON beamline. Each section describes the ex-situ calibrations used to correlate the concentrations of species in the electrolyte with neutron attenuations, and the characterization of concentration profiles in the *operando* flow cells under various voltage biases and flow configurations. In the NEUTRA section, two sets of experiments are discussed, one with a low attenuating supporting salt ($KPF_6$) and one with a highly attenuating counter-ion ($BF_4^-$), to differentiate between the redox active species and supporting ions.

### White beam neutron imaging (NEUTRA)

**Attenuation of electrolyte species.** Achieving contrast between the electrolyte constituents (solvent, redox-active species and supporting ions) is critical to identify species and quantify their dynamics within the electrochemical cell. White beam neutron imaging does not technically allow selectivity towards a target component, but it is possible to obtain insights into concentration distributions of individual electrolyte species by careful selection of the redox active species and supporting salt, coupled with the subtraction of reference images. We capitalize on the flexibility in the choice of solvent, supporting electrolytes and redox-active molecule for NAqRFBs, and measure attenuation coefficients for a set of electrolyte types and components using cuvettes (Fig. 2a). The attenuation difference between the pure deuterated solvent ($CD_3CN$) and 0.2 M supporting salt solution ($KPF_6$ in $CD_3CN$) is sufficiently small to be neglected, confirming the negligible attenuation of $KPF_6$ at this neutron energy and concentration. On the other hand, the addition of 0.5 M TEMPO in this electrolyte solution results in an increased attenuation coefficient as it has four methyl groups rich in hydrogen atoms attached to a piperidine ring (molecular formula $C_9H_{18}NO \cdot$). The large number of hydrogen atoms results in a stark contrast between the supporting salt ($KPF_6$) and the active species (TEMPO/TEMPO⁺). For the concentration range investigated in this study (0−0.5 M), TEMPO and TEMPO⁺$PF_6^-$ dissolved in $CD_3CN$ show similar neutron attenuations (Fig. 2a, b). The similar cross sections of TEMPO and TEMPO⁺ are expected given their identical chemical composition (only one electron difference) resulting in almost identical interactions with neutrons, and further confirms the low attenuation of $PF_6^-$ ions. Finally, when the counter-ion ($PF_6^-$) of TEMPO⁺ is replaced with $BF_4^-$, the solvent-corrected attenuation at the same concentration is nearly doubled (Fig. 2a), which indicates that TEMPO species and $BF_4^-$ ions have similar microscopic cross-sections. Although the counter-ion contains no hydrogen atoms, $BF_4^-$ contains boron which features a large neutron absorption cross-section for thermal neutrons[53]. Figure 2b shows a linear correlation between neutron attenuation *vs.* concentration for the different species employed in this study, which confirms the validity of the chosen concentration range (0−0.5 M) where the Lambert-Beer law (Eq. (1)) holds. These reference measurements are then used to obtain local concentrations in the electrochemical reactor volume during operation.

**Transport of the active species.** We performed neutron imaging on an *operando* redox flow cell to visualize concentration profiles of TEMPO/TEMPO⁺ (Fig. 3). The cell is connected to tanks with 50% SoC TEMPO/TEMPO⁺ at 0.5 M concentration on the counter electrode (CE) side and 0.2 M on the working electrode (WE) side, both with 0.2 M $KPF_6$ to provide ionic conductivity and minimize supporting salt impact on neutron attenuation (Fig. 3a). We chose to have an offset in concentrations between both compartments to study the diffusive flux in the absence of reactions. Because the WE and CE compartments are separated by an anion exchange membrane, the transport of cations such as TEMPO⁺ and K⁺ is significantly hindered, whereas the anions and neutral molecules such as $PF_6^-$ and TEMPO can more easily pass through. Furthermore, we elect to use a parallel (flow-by) flow field to limit the convective transport through the porous electrodes. Using this cell architecture and due to the negligible neutron attenuation of $KPF_6$, we can track the movement of TEMPO between the electrodes. The cell is discharged (negative potential applied at the WE) and charged (positive potential applied at the WE) alternately, such that the state-of-charge after each complete cycle does not significantly deviate from the initial condition and two voltage magnitudes were applied to understand their impact on the potential-driven transport processes (e.g., migration). The electrochemical sequence goes

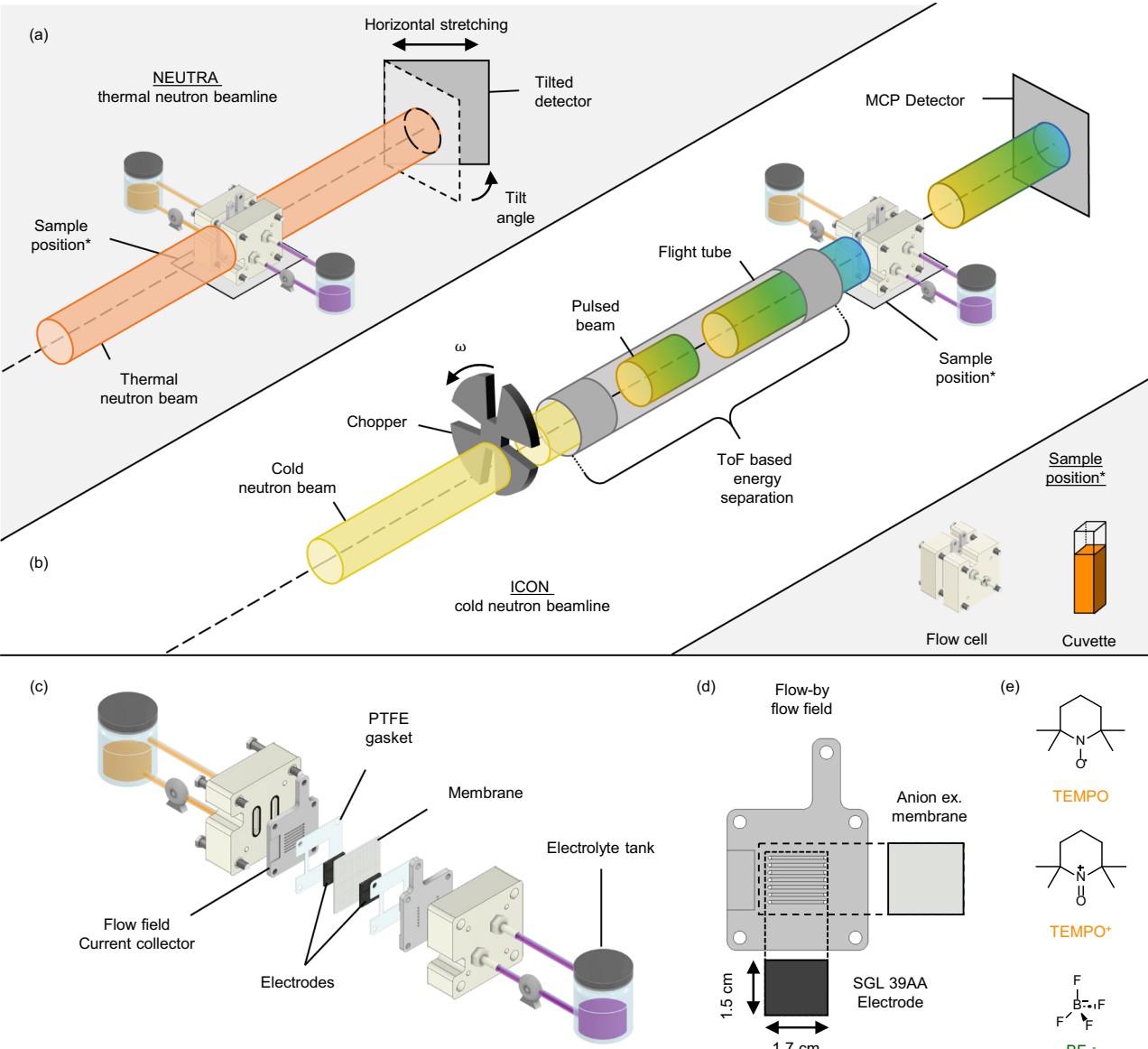

**Fig. 1 | Illustration of the neutron radiography approach showing the neutron imaging set-ups and cell components.** Schematic representations of (**a**, **b**) the neutron imaging set-ups, and (**c**–**e**) the flow battery cell design and components, utilizing non-aqueous electrolytes. **a** Neutron imaging using the NEUTRA beamline, where a cumulative concentration profile of TEMPO, TEMPO$^+$ and BF$_4^-$ species is obtained. **b** Neutron imaging using the ICON beamline, where the moderation of the neutron velocities enables to select the energy of the neutrons by means of a mechanical chopper disk, a flight tube and a microchannel plate detector, where deconvoluted concentration profiles of active species (TEMPO and TEMPO$^+$) and supporting electrolyte anions (BF$_4^-$) are obtained. The cell positions in **a**, **b** were used for both flow cell and cuvette imaging. **c** Exploded view of the flow cell components and the connected electrolyte tanks. **d** Representation of the cell components (parallel flow field, anion exchange membrane and SGL 39AA porous electrode) with their dimensions. **e** Chemical structures of the attenuating active species TEMPO and TEMPO$^+$, and the attenuating supporting species BF$_4^-$.

through the open circuit voltage (OCV), −0.3, +0.3, −0.6 and +0.6 V steps, each for 20 min at the highest tested inlet flow rate of 15.1 mL min$^{-1}$ (Fig. 3b). We also studied the impact of flow rate by performing the same electrochemical sequence (without the OCV step) at 5.6 mL min$^{-1}$ (see Figure S1 in the Supporting Information). The current-time and voltage-time curves of the entire experiment can be found in Figure S2 and a video of the experiment can be found in the Supplementary Materials (see Supplementary Video 1). *Operando* imaging of the cell during the electrochemical protocol results in transmission images where the attenuation at each location represents the integral of neutron-matter interactions along the neutron path (Fig. 1a). These images are then averaged for the duration of a voltage step (20 min) and result in concentration maps for a given condition at steady-state (Fig. 3c, d). The colour scale represents the cumulative concentration

of TEMPO and TEMPO$^+$ and ranges from 0-0.5 M, resulting in a 2D map of the species concentration in the reactor area. The membrane area is omitted as the quantification of concentrations is not reliable in this region due to the high hydrogen content of the polymer membrane (perfluorinated with a polyketone reinforcement) and the reduced membrane thickness (130 μm). Finally, we calculate the concentration profiles across the thickness of the electrodes and compute these between the flow field-electrode interfaces of both half cells. Using this approach, one-dimensional concentration profiles, parallel with the membrane plane, are obtained.

The experiment begins with an OCV step where no current is drawn from the cell. The brighter colour of the CE side in the OCV radiograph represents a higher total active species concentration compared to the WE side, as expected by the concentrations of the

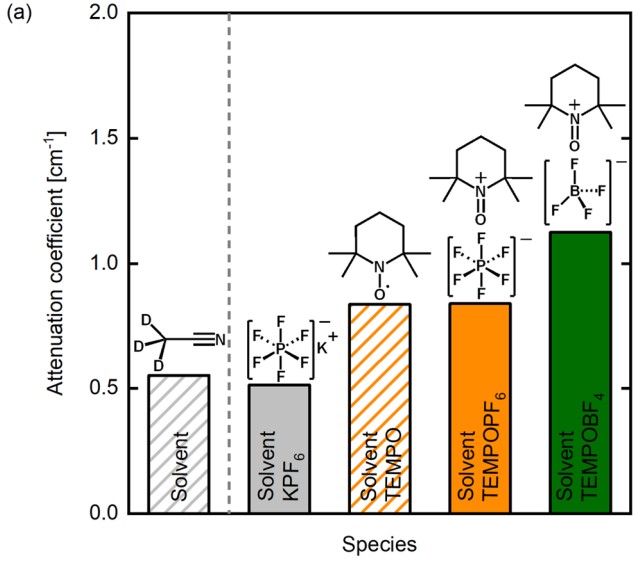

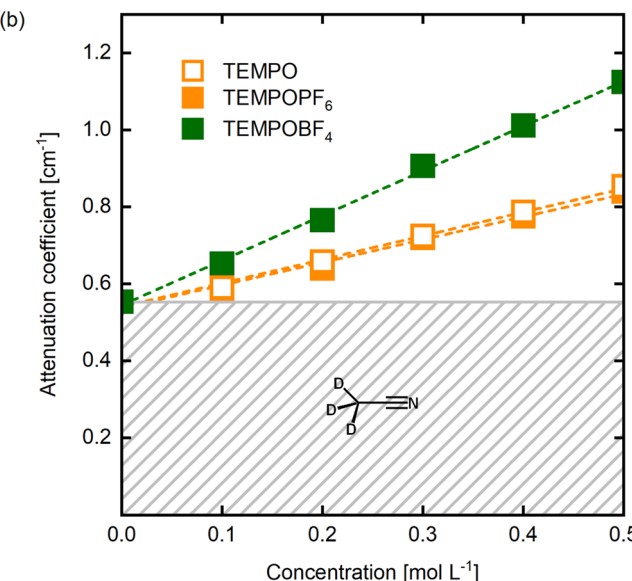

**Fig. 2 | Determination of the attenuation coefficient for the chemicals used in this study obtained at the NEUTRA beamline. a** The attenuation coefficients of the different species in $CD_3CN$: the solvent only, supporting electrolyte (0.2 M) and species TEMPO, TEMPO$^+$PF$_6^-$ and TEMPO$^+$BF$_4^-$ (all 0.5 M). **b** The linear dependence of the attenuation coefficient of the TEMPO, TEMPO$^+$PF$_6^-$ and TEMPO$^+$BF$_4^-$ species on the concentration (0.1, 0.2, 0.3, 0.4 and 0.5 M), where the shaded area represents the attenuation of the solvent.

electrolyte fed (0.5 M and 0.2 M TEMPO/TEMPO$^+$). An advantage of neutron radiography is that electrolyte wetting of the porous electrodes can be visualized because of the low attenuation of gasses, which will appear as dark spots (i.e., lower concentration) in the radiographs[54]. During the OCV period, the concentration on both sides does not show dark regions (Fig. 3c, d), suggesting full wetting, at least to the spatial resolution of the measurement. Moreover, over the course of the OCV period, the concentration profile remained fairly constant which can be attributed to the low diffusion rate of TEMPO/TEMPO$^+$ through the dense anion exchange membrane. Overall, the concentration profiles under cell polarization do not strongly deviate from the initial OCV state, except at positive potentials near the membrane area on the WE. The OCV profile can be explained by the reactor configuration (i.e., anion exchange membranes, flow-by flows fields) and the low ionic conductivity of the nonaqueous electrolyte,

resulting in low current densities and charge consumption (< 15% of the tank capacity, see Figure S3). In this experiment, the concentration gradient is from left to right due to the higher cumulative TEMPO/ TEMPO$^+$ concentration on the CE side. Under negative potentials (−0.3 V and −0.6 V), TEMPO is converted to TEMPO$^+$ in the CE side, resulting in a build-up of TEMPO$^+$, while the opposite reaction is taking place in the WE, resulting in TEMPO$^+$ depletion. To compensate for the charge, PF$_6^-$ crosses through the membrane towards the CE side, which is not visible in the images due to its low attenuation. Although favourable to sustain the electrochemical reaction, we do not expect TEMPO to cross to the CE side on the timescale of this experiment as this would be against its concentration gradient, thus the images and profiles for negative potentials are nearly identical to the OCV conditions (Fig. 3c).

On the contrary, when positive potentials are applied (+ 0.3 V) a concentration gradient develops in the WE and does not return fully to the OCV concentration profile when a positive potential step is applied afterwards. This result is supported by the bright concentration front in the corresponding radiographs near the membrane and could be attributed to the accumulation of TEMPO+ at the membrane due do Donan exclusion. Increasing the potential to +0.6 V amplifies this trend, exacerbating the concentration gradient and extending the concentration front deeper within the WE (Fig. 3d). Decreasing the flow rate to 5.6 mL min$^{-1}$ further intensifies the concentration front in the WE (Figure S1) for both the positive and negative potentials. Interestingly, this concentration front does not appear in the CE at negative potentials, motivating the use of energy-selective neutron radiography to deconvolute the concentrations of the different species. The concentration front reveals mass transfer limitations, determined by the membrane properties, applied potential, electrolyte velocity, species concentration and electrolyte and electrode properties[8]. To visualize such limiting phenomena, we utilized a flow-by flow field design that induces limited convection within the porous electrode. However, the intensification of the concentration fronts at lower flow rate suggests that this flow field does have convective transport contributions in the electrode. Nevertheless, we anticipate that a convection-enhanced flow field (such as interdigitated or flow-through) would further increase species replenishment and reduce concentration gradients[55]. In this first set of experiments, the low attenuation coefficient of the KPF$_6$ salt was utilized to maximize the contrast of TEMPO and TEMPO$^+$ compared to other electrolyte components. To visualize the motion of anions, we then employ a strongly attenuating counter-ion (BF$_4^-$ instead of PF$_6^-$) without any additional supporting salt to amplify the contrast between all species in the electrolyte.

**Transport of the counter-ion.** Supporting ions are essential in RFBs to provide ionic conductivity[56,57]. Here we leverage BF$_4^-$ as counterion due to its high neutron attenuation (see *Attenuation of electrolyte species*)[58]. To assess the influence of migration on the charged species transport (i.e., stoichiometric operation), we do not add a supporting salt in the electrolyte (Fig. 4a), which negatively impacts the obtained current density (Figure S4) but enables visualization of the counterion. To amplify the counterion concentration, we utilize 0.5 M TEMPO on the CE side and 0.5 M TEMPO$^+$BF$_4^-$ on the WE side. Furthermore, the use of an anion-exchange membrane significantly restricts TEMPO$^+$ movement between compartments, which leaves BF$_4^-$ as the main charge carrier. As we perform subtractive neutron imaging, the isolation of [TEMPO], [TEMPO$^+$] and [BF$_4^-$] is not possible with white beam neutron imaging, resulting in cumulative concentration maps (see the combined colour scale in Fig. 4). Although the concentration information is cumulative, by tuning experimental parameters (type of ion-exchange membrane and tank concentrations) we hypothesize that the observed changes can be attributed to the motion of certain species, which is

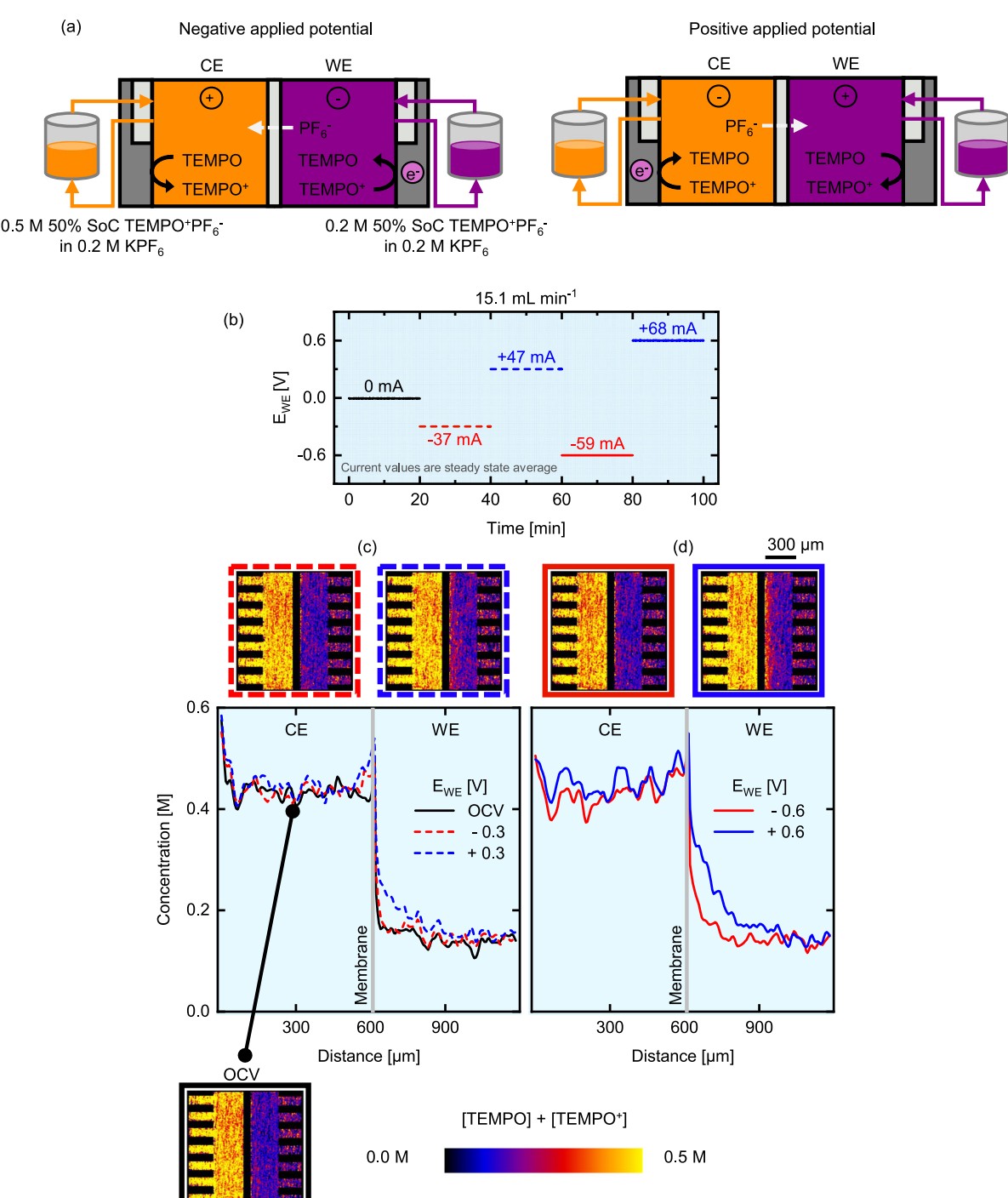

**Fig. 3 | *Operando* imaging of the active species transport in the NEUTRA beamline with the low attenuating KPF₆ supporting salt. a** Schematic representation of the non-aqueous cell designs during charge and discharge mode, where the counter electrode (CE) corresponds to 0.5 M TEMPO/TEMPO⁺PF₆⁻ at 50% state-of-charge in 0.2 M KPF₆ and the working electrode (WE) to 0.2 M TEMPO/TEMPO⁺PF₆⁻ at 50% state-of-charge in 0.2 M KPF₆. **b** Electrochemical sequence over time showing the applied potential steps and measured averaged current output at an inlet flow rate of 15.1 mL min⁻¹. **c, d** Cumulative active species (TEMPO/TEMPO⁺) concentration profiles over the electrode thickness at an inlet flow rate of 15.1 mL min⁻¹. The averaged snapshots of the cell after image processing and the concentration profiles are shown for various applied potential steps: **c** OCV, −0.3 V and +0.3 V and **d** −0.6 V and +0.6 V.

predominantly BF₄⁻ in this configuration. This resonates with the subtle changes in the concentration profiles in Fig. 3 as a function of the applied potential, hinting that the neutron-transparent PF₆⁻ is the main charge carrier in the system. A novel approach to isolate the concentration of species in solution with neutron imaging is discussed in the *energy selective imaging* section.

In the first hour of the experiment, the system is kept at OCV conditions to track diffusional crossover through the membrane (Fig. 4b). Although we track a change in OCV over time, indicating crossover and concentration equilibration, the small concentration variations in this short time period are not captured by the radiographs (OCV radiograph in Fig. 4c). In this experiment, BF₄⁻ has a strong concentration gradient towards the CE side, however the OCV profile does not show significant deviation from the initial concentrations, which is attributed to the Donnan exclusion of TEMPO⁺ coupled with the barrier properties of the dense anion exchange membrane[59].

Furthermore, the concentrations in the reactor volume are homogeneous and no local fluctuations are observed. We conclude that the timeframe of the OCV period is shorter than the time needed for diffusional crossover of species for this configuration.

After the OCV period, alternating potential steps are applied to the electrochemical cell and the current response is recorded in time (Figure S4). A video sequence of the experiment can be seen in Supplementary Video 2. At negative potentials, TEMPO$^+$ is converted to TEMPO in the WE, and BF$_4^-$ migrates to the CE compartment (Fig. 4a). We can observe a local accumulation of attenuating species in the vicinity of the membrane on the CE side, while the opposite trend is observed in the WE compartment (Fig. 4c), attributed to the migration of the BF$_4^-$ anion from the WE to the CE to maintain electroneutrality. This effect is more pronounced at a higher potential (−0.6 V *vs.* −0.3 V, Fig. 4c), which illustrates the influence of migration. A starker concentration gradient between compartments is obtained at lower flow rates (5.6 mL min$^{-1}$, Fig. 4d), as the convective mass transfer is lower. At the WE, we find higher concentrations in the areas near the flow field inlets in comparison with the area under the ribs, showing an advantage of two-dimensional concentration maps obtained using neutron radiography. We hypothesize that velocity distributions within the porous electrodes, induced by the flow-by-flow field design and the relatively thick porous electrode, explain these variations in concentration through the electrode volume. This phenomenon is more visible at the highest flow rate as the convective forces pushing the electrolyte in the porous electrode are larger (Fig. 4d). In summary, at negative potentials, the profiles show an increasing species concentration in the CE occurring synchronously with a decrease in the WE (Fig. 4c).

When a positive potential is applied to the WE, the reverse reactions take place, and the resulting radiographs and the concentration profiles (Fig. 4e) are closer to the initial OCV profile compared to negative applied potentials. Because of the reactor architecture used in the NEUTRA experiments (stacked paper electrodes and a flow-by-flow field) in combination with the low ionic conductivity of the electrolyte, there is only a small change in the cell capacity at negative potentials, consuming only ~8% of the total capacity (Figure S5), resulting in relatively low current densities. Therefore, at positive potentials, only a small amount of TEMPO is present in the WE compartment to be converted back to TEMPO$^+$. As a result, large overpotentials are generated throughout the cell due to the low concentration of reactants to sustain the current. This explains the asymmetry in the current magnitudes when the polarity of the cell is reversed (i.e., +7 mA vs. −65 mA at 15.1 mL min$^{-1}$ and +/−0.3 V, Fig. 4b), and the even lower capacity recovery (~1-2%) resulting in underutilized capacity over the duration of the experiment (Figure S5).

When comparing the experiments with counter-ion BF$_4^-$ and supporting salt KPF$_6$, we can correlate the macroscopic performance with the concentration distributions through the reactor. For the KPF$_6$ experiments, all active species (i.e., TEMPO, TEMPO$^+$, PF$_6^-$) are present in both compartments. Therefore, the macroscopic performance, i.e., the current output, is symmetric when operating at negative and positive applied potentials (Fig. 3b) as the to-be-reacted species are present without the requirement of species crossover under the evaluated conditions (as the capacity change is limited to ~20%). The symmetric current output results in concentration profiles returning to the OCV profile when positive potentials are applied. Whereas for the BF$_4^-$ experiment, the charged species are only present in one compartment (WE) initially and are required to cross the membrane to support the reactions, which is limited by the anion exchange membrane, resulting in asymmetric current magnitudes upon changing cell polarities (Fig. 4b). The asymmetric current can be correlated to the concentration profiles, as the concentration

does not fully return to the OCV profiles for positive applied potentials (Figure S6).

Using white beam neutron imaging, we have obtained cumulative concentration maps which include active species and supporting electrolytes. Using this approach, we have coupled macroscopic electrochemical cell performance with microscopic concentration distributions, revealing mass transfer modes under different cell potentials, flow rates and cell polarities. However, we are not able to isolate concentrations of active species and supporting ions with this incident beam. Acknowledging these limitations, we then utilize time-of-flight imaging to visualize reactive transport phenomena of both the active species and counter-ion, under similar experimental conditions.

## Energy-resolved neutron imaging (ICON)

In pursuit of deconvoluting the concentrations of different species in the electrolyte, we investigate the use of energy-resolved neutron radiography at the ICON beamline. This beamline utilizes a colder neutron spectrum by secondary moderation of the neutron beam, and slower neutrons undergo inelastic scattering events with a higher probability than thermal neutrons, allowing more variations in species cross-sections to be observed. It is also possible to perform spectral neutron imaging with a time-of-flight based technique at ICON, which is currently not possible at the NEUTRA beamline due to space limitations. Since the time-of-flight of neutrons in the flight tube is inversely proportional to the square root of their energy, the ToF-NI technique can add a fourth dimension to conventional radiography. This can provide an additional mode of contrast as neutron attenuation is a function of its energy. We anticipate that if the neutron attenuation of active species and the supporting ions have distinct energy dependency profiles, we can separate the contribution of each species from the final radiograph.

**Correlating attenuation with neutron energy.** The difference in relative neutron attenuation of materials enables tuning of the contrast between different species. To this end, we first performed calibration experiments with cuvettes, filled with 0.5 M solutions of TEMPO and TEMPO$^+$BF$_4^-$ in CD$_3$CN. Figure 5a shows attenuation coefficients as a function of the time of flight, where the BF$_4^-$ attenuation coefficient is determined by subtracting the coefficient of TEMPO from TEMPO$^+$BF$_4^-$. Here, an increasing time-of-flight indicates a decreasing neutron energy. TEMPO$^+$BF$_4^-$ reaches nearly twice the cross-section of TEMPO at higher energies, corroborating the previous observations made at the NEUTRA beamline (Fig. 2) that TEMPO and BF$_4^-$ have similar microscopic cross-sections. The linearity of the concentration with neutron attenuation was already demonstrated in the NEUTRA beamline, thus we selected only one concentration (0.5 M) corresponding to the starting concentration in the flow cell experiments.

Using the matrix operation shown in Eq. (4) and in Fig. 5b, the respective contributions of TEMPO and BF$_4^-$ from the total neutron attenuation can be separated. For this purpose, we need to define two regions within the spectrum, the high energy (HE) and low energy (LE) regions (see the *Neutron Radiography* section). The difference in attenuation coefficients between TEMPO and BF$_4^-$ varies as a function of neutron energy, this means that the slope of the graph in Fig. 5a should be different between species, or in mathematical terms, the determinant of the microscopic cross-section matrix should not be zero. The microscopic cross-sections of the species of interest are reported for HE and LE regions in Fig. 5b. The values reported here correspond to the microscopic cross-section averaged over LE and HE ranges, described in the *Neutron Radiography* section. Although maximum contrast is achieved around 8 ms ToF, the LE region was moved towards higher energies to prevent the excessive neutron edge effects/scattering at interfaces between gaskets observed at lower energies. Finally, the matrix operation is applied pixel-wise to the

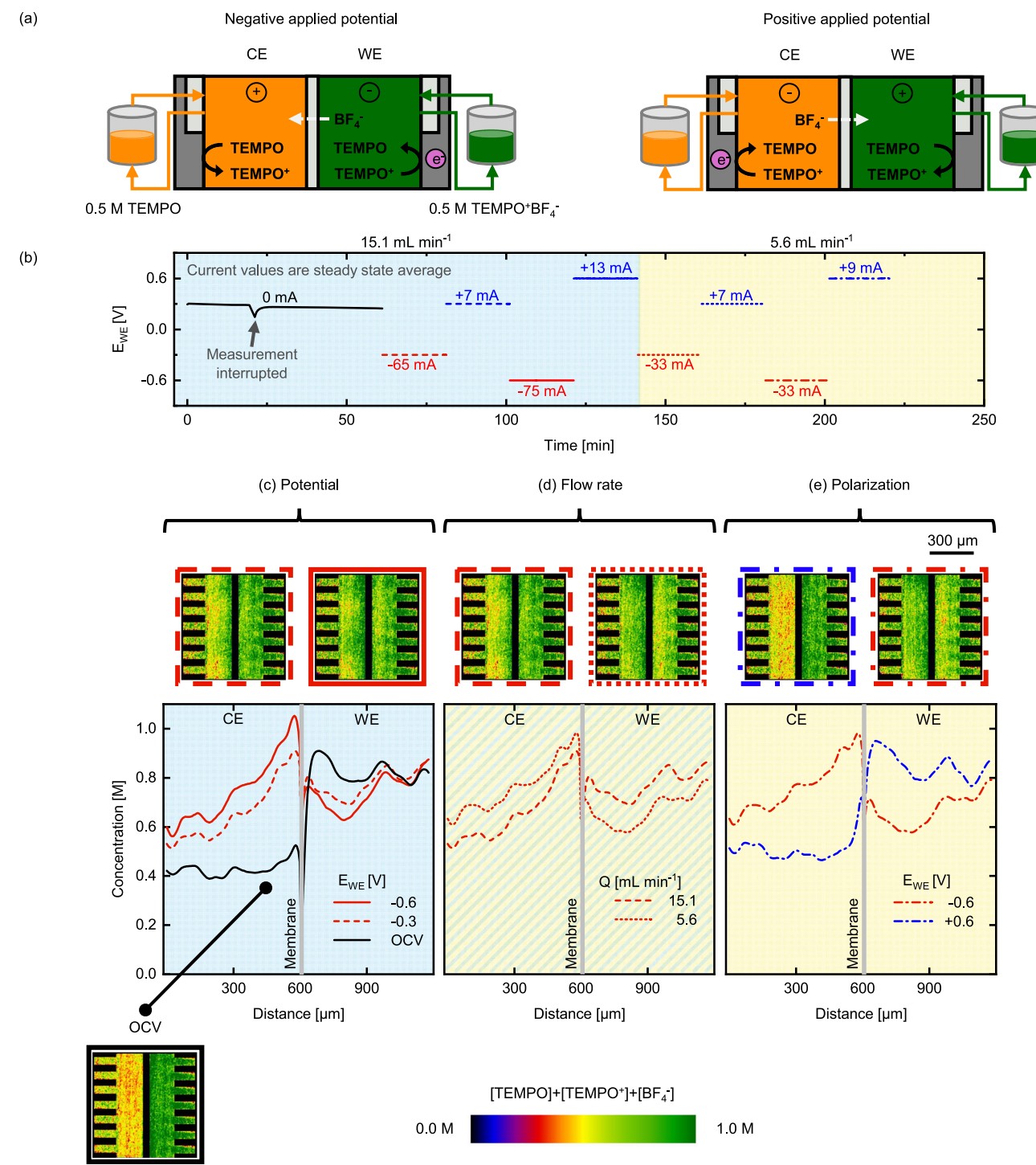

**Fig. 4 | *Operando* imaging of the active species transport in the NEUTRA beamline with the neutron attenuating BF$_4^-$ supporting ion. a** Schematic representation of the non-aqueous cell designs during charge and discharge mode, where the counter electrode (CE) corresponds to 0.5 M TEMPO and the working electrode (WE) to 0.5 M TEMPO$^+$BF$_4^-$. **b** Electrochemical sequence over time showing the applied potential steps and measured averaged current output at two inlet flow rates of 15.1 mL min$^{-1}$ and 5.6 mL min$^{-1}$. **c–e** Cumulative active species

(TEMPO/TEMPO$^+$) and BF$_4^-$ supporting ion concentration profiles over the reactor area. The averaged snapshots over the whole period of each individual potential step of the cell after image processing and the concentration profiles are shown for various applied potential steps and show the influence of various operation parameters: **c** applied potential magnitude (OCV, −0.3 V and −0.6 V at 15.1 mL min$^{-1}$), **d** flow rate (−0.3 V at 15.1 mL min$^{-1}$ and 5.6 mL min$^{-1}$) and **e** polarization sign (−0.6 V and +0.6 V at 5.6 mL min$^{-1}$).

greyscale transmission image to calculate the contribution of species, and a colour map is applied to designate the concentrations (Fig. 5b). Achieving contrast between TEMPO and TEMPO$^+$ is still not possible, but because the movement of TEMPO$^+$ between compartments is mostly blocked by the anion exchange membrane, we can track the

movement of BF$_4^-$ and TEMPO species separately during battery operation.

**Deconvoluting concentrations in a flow cell.** To demonstrate the potential of energy-selective and *operando* neutron imaging, a flow cell

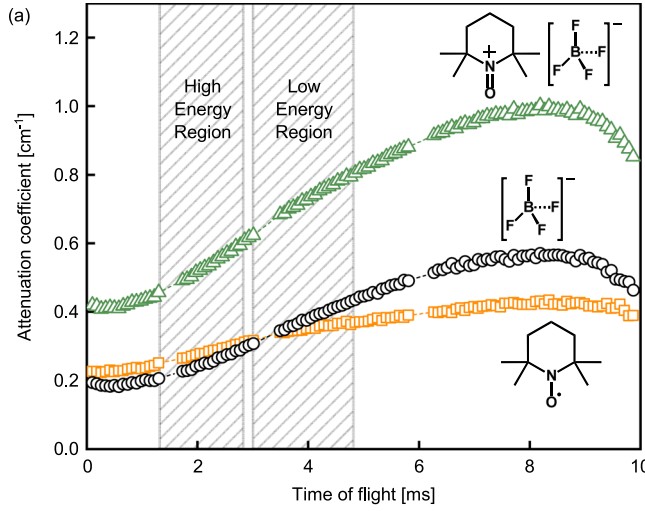

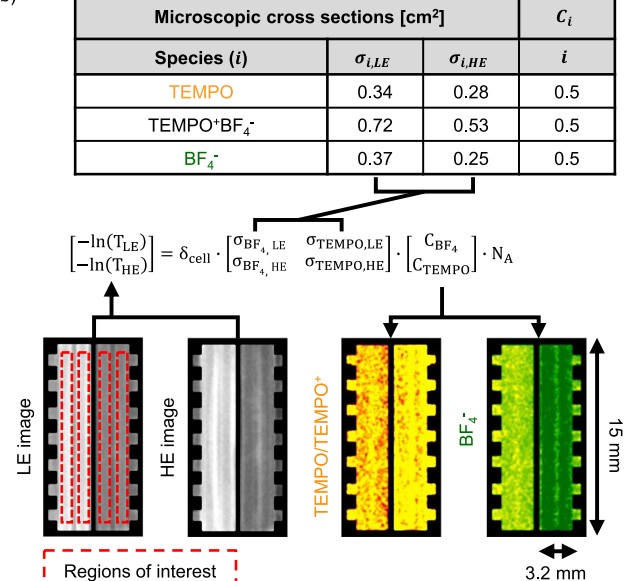

**Fig. 5 | Energy selective imaging at the ICON beamline. a** Energy dependency of the attenuation coefficient of TEMPO, TEMPO⁺BF₄⁻ and BF₄⁻ obtained from the cuvette experiments (all 0.5 M), where the time-of-flight is a function of the neutron energy. **b** Schematic representation of the main components in the image processing sequence for the ICON beamline experiments, including the table of microscopic cross-sections, the low and high energy transmission images (grayscale) with the regions of interest and deconvoluted active species and supporting ion images of the flow cell, showing the flow fields, electrodes and membrane together with their dimensions. The transmission images are processed using the given equation to extract the concentration maps (coloured images) of TEMPO/TEMPO⁺ and BF₄⁻.

with asymmetric concentrations (0.5 M TEMPO on the CE and 0.5 M TEMPO⁺BF₄⁻ on the WE side, Fig. 6a) was imaged. The electrolyte compositions are identical to the previous experiment that utilized BF₄⁻ as the counter-ion but we increased the electrode thickness to accommodate for the lower spatial resolution. Previous experiments carried out on the NEUTRA beamline were set up with a tilted detector to increase the spatial resolution of the images[60], resulting in a pixel size of ~6 μm applied to study a 630 μm thick electrode. In the ICON experiments, the ToF detection system resulted in a larger pixel size (~55 μm). Thus, to compensate for the discrepancies in spatial resolution, a thicker felt electrode (3200 μm) was employed. The cell features stacked gaskets (incompressible PTFE and compressible ePTFE) to enclose the thick felt electrode, where the interface around the ePTFE

gaskets shows up in the LE image as low transmission regions (dark vertical lines in the grayscale image in Fig. 5b) due to higher neutron edge effect/scattering. Nevertheless, the central regions of the incompressible gaskets (~1 mm) were large enough to define four regions of interest (Fig. 5b) where the concentrations can be determined, and the reported concentrations are averaged over this volume for both compartments. To track the movement of species during the sequence, we opted for plotting the averaged concentrations over time (Fig. 6c).

The experiment starts with an OCV period of 10 min, after which the cell was polarized at an electrolyte flow rate of 21.1 mL min⁻¹ followed by a reduced flow rate of 6.7 mL min⁻¹ (Fig. 6b and S7). From the deconvoluted concentration maps (Fig. 6c), we confirm that BF₄⁻ is the main charge carrier and that the membrane blocks the transport of TEMPO/TEMPO⁺, as their concentration remains relatively stable in both compartments throughout the entire electrochemical sequence. When a negative potential is applied, TEMPO⁺ reduces to TEMPO in the WE compartment while the reverse reaction occurs in the CE (Fig. 6c). Simultaneously, the concentration maps and averaged concentrations show that BF₄⁻ moves through the membrane towards the CE to balance the positive charge of the generated TEMPO⁺ species. Applying a positive potential to the WE reverses the direction of the migration flux of the BF₄⁻ ions, and concentrations close to the initial state of the battery (i.e., OCV) can be recovered. These results corroborate the observations from the experiments in NEUTRA as only minor concentration fluctuations were observed with PF₆⁻ as supporting salt when an electric field is applied, whereas stark changes are detected when the supporting ion was changed to BF₄⁻ due to its strong neutron attenuation. Moreover, the concentrations of the active species within the reactor area show larger variations for the highest flow rate (Fig. 6c), induced by faster species conversion (i.e., higher current densities, Fig. 6b), and greater convective transport in the porous electrode. This brings the concentrations to extreme values due to the fast depletion of reactants in the electrolyte and promotes larger ionic currents. The difference in ionic current from the electrochemical data is correlated to the slope of the BF₄⁻ concentration variations as a function of time.

From the capacity curves (Figure S8), we observe that after the first potential step (−0.6 V at 21.1 mL min⁻¹), 60% of the total capacity is discharged. In the next step, after an applied potential of +0.6 V, only 45% of the total capacity is recharged (due to the lower current at set time), resulting in 15% underutilized capacity after a full polarization cycle because of the starting tank solutions (no BF₄⁻ in the CE) as explained in the *transport of the counter-ion* section. At the lower flow rate (6.7 mL min⁻¹), the capacity consumed at negative applied potentials is almost fully recovered at positive applied potentials, resulting in near symmetric current magnitudes. Interestingly, we find comparatively higher currents and capacity utilization with this reactor configuration in comparison with the reactor architecture used in the NEUTRA beamline (Fig. 4), which can be correlated to the significant BF₄⁻ concentration fluctuations. We attribute these differences to the use of a different electrode material (a thick felt *vs.* a stack of thin carbon papers) and lower compressive forces. The higher porosity, apparent permeability and internal surface area of the felt electrode can explain the higher current densities observed in this reactor configuration.

Here, we demonstrate the potential of the ToF-NI spectral technique to isolate and visualize concentration distributions of active and supporting species in redox flow cells. Compared to the use of conventional neutron radiography, the ToF method requires larger acquisition times and provides lower spatial resolution but enables the detection of neutron energies necessary to deconvolute species concentrations.

### Practical application of the neutron imaging method

This work demonstrates for the first time the use of neutron radiography to image concentrations of redox-active species and

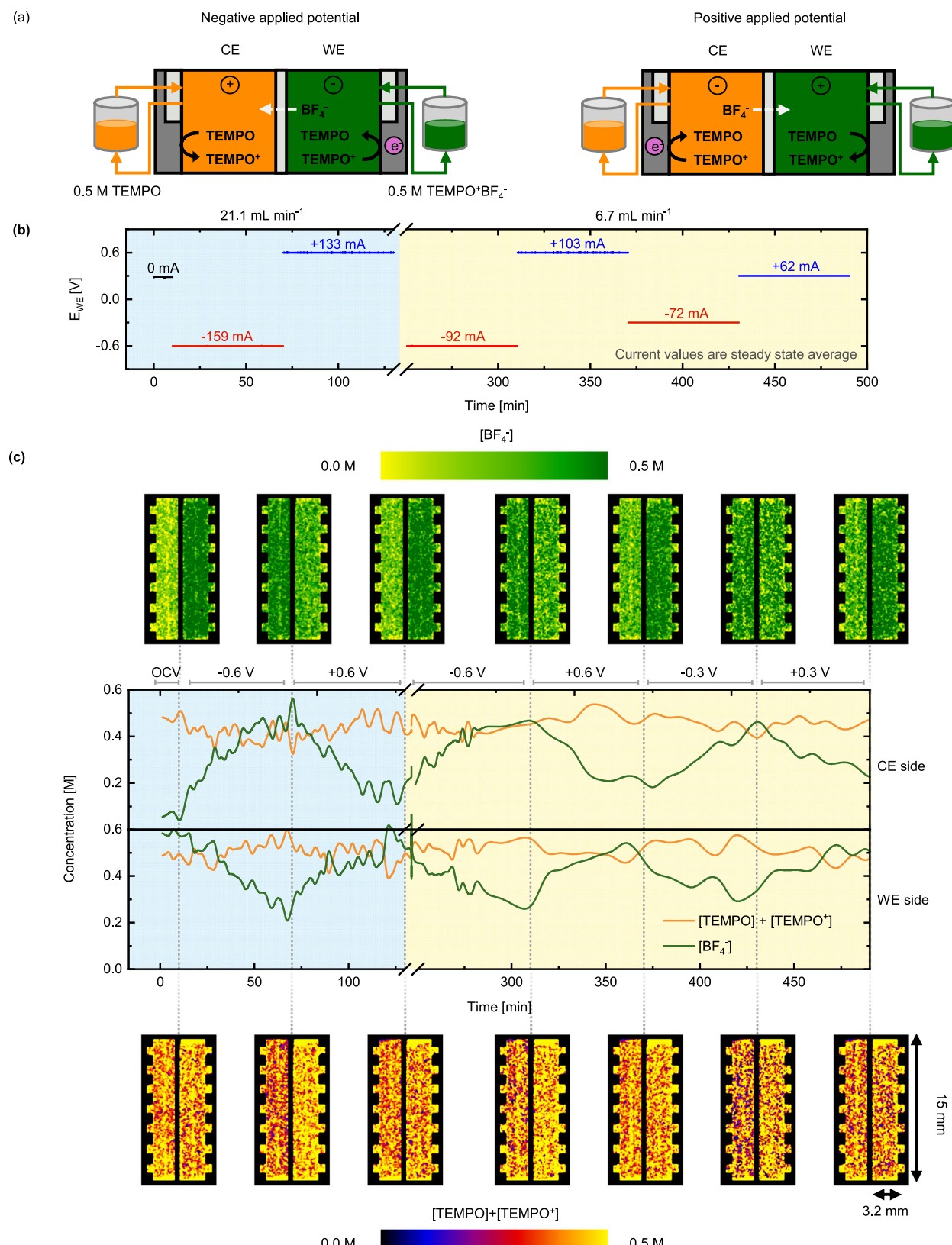

**Fig. 6 | *Operando* imaging of the active species transport in the ICON beamline with the neutron attenuating $BF_4^-$ supporting ion. a** Schematic representation of the non-aqueous cell designs during charge and discharge mode, where the counter electrode (CE) corresponds to 0.5 M TEMPO and the working electrode (WE) to 0.5 M $TEMPO^+BF_4^-$. **b** Electrochemical sequence over time showing the applied potential steps and measured averaged current output at two inlet flow rates of 21.1 mL min$^{-1}$ and 6.7 mL min$^{-1}$. **c** Deconvoluted active species (TEMPO/TEMPO$^+$) and $BF_4^-$ supporting ion concentration profiles. The averaged snapshots of the cell after image processing and the concentration profiles over time are shown for various applied potential steps and flow rates: OCV, −0.6 V and +0.6 V at 21.1 mL min$^{-1}$ and −0.6 V, +0.6 V, −0.3 V and +0.3 V at 6.7 mL min$^{-1}$, from left to right, where the OCV images are averaged over 5 images and the applied potentials averaged over 4 images.

supporting salts in *operando* electrochemical flow cells. By combining macroscopic electrochemical responses with microscopic concentration distributions, neutron radiography can provide valuable insights into species motion within the reactor area, and this can be used to quantify mass transport methods (migration, diffusion, convection) and phenomena affecting the performance of the battery during operation (e.g., electrolyte depletion, precipitation, physical failure in RFB stacks). These insights can be directly used to compare and select optimal cell components and to aid computational efforts. We anticipate that the use of molecular engineering to design redox molecular probes with controlled molecular structure, diffusivity and redox potential, can enable deconvolution of different oxidation states and degradation products. Although we focus on nonaqueous redox flow cells as a case study, neutron imaging can be extended to numerous applications to quantify concentration distributions in electrochemical cells and beyond. First, the resulting concentration maps can be used as experimental data to validate computational models that describe reactive mass transport. Here we used an electrolyte composed of a solvent, supporting electrolyte, and two redox active species - hence a complex, multicomponent system close to practical devices; but model experiments can be performed (e.g. with one or two analytes) to systematically deconvolute mass transport modes (diffusion, convection and migration) and their associated transport rates. Deeper fundamental understanding of reactive mass transport in electrochemical reactors and through membranes can aid in the design of advanced electrochemical cells. Second, the methodology enables identification of local maldistributions in concentration, which can assist in designing better flow field geometries and electrodes, as well membrane crossover, Donnan exclusion and salt precipitation within the electrochemical cell, which are deleterious to performance and lifetime in several flow battery chemistries (e.g. non-aqueous, all-vanadium, all-iron). Third, we anticipate that the method - and adaptations on the detection physics - will be instrumental in advancing hybrid redox flow batteries (e.g. all-iron, zinc-bromine), where there are phase change reactions (e.g. plating and stripping or hydrogen evolution) that fundamentally limit the performance of the system. Fourth, we expect that the technique can be applied to technical systems such as electrochemical stacks, where traditional neutron imaging was instrumental in the development of fuel cell stacks through visualization of water distributions. Fifth, beyond redox flow batteries, the method can be applied to other (electro)chemical reactors where concentration profiles determines performance such as electrochemical separations, flow chemistry and chemical reactor design. To further assist the design of neutron experiments to study electrochemical systems, Tables S1 and S2 summarize the neutron attenuation coefficient of commonly used materials for reactor manufacturing and redox species/supporting salts, respectively.

Finally, the use of molecularly engineered redox molecules acting as imaging probes might enable simultaneous visualization of the concentration of multiple components (> 2) when combined with energy-selective neutron imaging. For example, minimizing the neutron attenuation of the redox active molecules through a reduced hydrogen content or deuterium labelling is a viable strategy to obtain contrast. Although the technique is still in its early stages, it displays considerable potential. We anticipate that ongoing advancements in neutron detectors and choppers will enable more sophisticated analyses of complex multicomponent systems using ToF-NI, offering enhanced spatial, temporal and energy resolution.

## Methods

### Chemicals
2,2,6,6-Tetramethylpiperidin-1-yl)oxy (TEMPO, Sigma Aldrich, 98 %), nitrosonium tetrafluoroborate (NOBF$_4$, Thermo Scientific, 98%), nitrosonium hexafluorophosphate (NOPF$_6$, Thermo Scientific, 95%), acetonitrile-d3 (CD$_3$CN, Zeochem AG, 99.8%D), acetonitrile (CH$_3$CN, Sigma Aldrich, ≥ 99.9%) and potassium hexafluorophosphate (KPF$_6$, Thermo Scientific, 99%) were used without further purification.

### Electrolyte preparation
TEMPO was converted to its cation form 2,2,6,6-Tetramethyl-1-piperidinyloxy-oxo (TEMPO$^+$) via chemical oxidation with nitrosonium salts in a nitrogen-filled glove box (MBraun, LABstar, O$_2$ < 1 ppm, H$_2$O < 1 ppm). TEMPO (12.52 g) was dissolved in acetonitrile, where 1.1 molar equivalents of nitrosonium hexafluorophosphate (NOPF$_6$, 15.42 g) or nitrosonium tetrafluoroborate (NOBF$_4$, 10.29 g) dissolved in CH$_3$CN were slowly added during 2 hours to prevent NO$_x$ build-up[12]. Then, a rotary evaporator (40 °C, gradual decrease from atmospheric pressure to vacuum) was used to remove the solvent and the TEMPO$^+$PF$_6^-$ or TEMPO$^+$BF$_4^-$ salts were recovered. The electrolytes were prepared by weighting the solid fractions (TEMPO, TEMPO$^+$PF$_6^-$ and KPF$_6$ or TEMPO and TEMPO$^+$BF$_4$) prior to adding CD$_3$CN until full dissolution. Finally, the volume was adjusted in a graduated flask to reach 20 mL of total volume. For the experiments using KPF$_6$ as supporting salt, a 50% state-of-charge (SoC) solution was prepared for each side with two different total concentrations of TEMPO species (0.5 M at the working electrode (WE) side and 0.2 M at the counter electrode (CE) side) in 0.1 M KPF$_6$. For the experiments using BF$_4^-$ ions, 0.5 M of TEMPO was used at the CE against 0.5 M of TEMPO$^+$BF$_4^-$ at the WE without additional added supporting salt to achieve high BF$_4^-$ concentrations without compromising the imaging process. For all imaging experiments, CD$_3$CN was used as the solvent to reduce its contribution to the overall transmission as deuterium has a 10-fold lower total cross-section than hydrogen (at a neutron velocity of 2200 m s$^{-1}$)[58].

### Calibration experiments
To quantify the concentration of the species within the electrochemical cells, the attenuation coefficients were determined with cuvette calibration measurements to correlate neutron transmission with solution concentration. The attenuation of the beam by the sample was calculated using the Lambert-Beer law[50], described as

$$T = e^{-\sigma_i n_i \delta} \tag{1}$$

where $T$ is the transmitted intensity after correction for the attenuation of an empty cuvette [-], $\sigma_i$ the conventional microscopic cross-section [m$^2$], $n_i$ the number density of species i [m$^{-3}$], which is related to the concentration of species i by $C_i = n_i/N_A$ [mol m$^{-3}$], and $\delta$ is the thickness of the cuvette [m].

For the NEUTRA and ICON beamline measurements, cuvettes (1 cm optical path) were filled with different electrolyte solutions to calibrate the attenuation of the neutron beam. For the NEUTRA beamline experiments, the reference cuvettes were filled with CD$_3$CN (solvent) or 0.2 M KPF$_6$ in CD$_3$CN (supporting electrolyte). To verify the linearity between concentration and neutron attenuation, solutions of TEMPO, TEMPO$^+$PF$_6^-$ and TEMPO$^+$BF$_4^-$ in CD$_3$CN were measured at molarities of 0.1, 0.2, 0.3, 0.4 and 0.5 M. For the ICON beamline measurements, a single concentration point method was used as the linearity of the neutron attenuation as a function of the electrolyte concentration was obtained at the NEUTRA beamline. The reference cuvette was filled with CD$_3$CN (solvent), and the single concentration points were measured for the solutions of interest of 0.5 M TEMPO and 0.5 M TEMPO$^+$BF$_4^-$ in CD$_3$CN.

### Flow cell parts
Neutron imaging was performed using a laboratory-scale redox flow cell with minimal modifications for imaging. The flow diffusers were machined from polypropylene (McMaster-Carr) and the graphite

parallel flow-by flow fields, also functioning as current collectors, featuring seven 1.6 cm long flow channels (0.5 mm depth and 1 mm width) were milled from 3.18 mm thick resin-impregnated graphite plates (G347B graphite, MWI, Inc.)[12]. All flow cells employed a Fumasep FAB-PK-130 (Fuel Cell Store, dry thickness 130 μm) anion exchange membrane. The electrodes had a geometric area of 2.55 cm$^2$ enclosed within incompressible polytetrafluoroethylene gaskets (ERIKS) and/or compressible polytetrafluoroethylene (ePTFE, Gore®, 520 μm nominal) gaskets to improve the sealing of the flow cell (Fig. 1c). For the experiments with the NEUTRA beamline, the flow cells contained three Sigracet 39AA electrodes (Fuel Cell Store, 280 μm nominal thickness, 89% porosity) per anode and cathode sides (six electrodes in total), sandwiching the membrane. The electrodes were compressed at 25% compression by selecting an incompressible gasket thickness of 630 μm. For the experiments at the ICON beamline, one AvCarb G100 felt electrode (Fuel Cell Store, 3200 μm nominal thickness, 95% porosity) per side was used. Each side was sealed with two incompressible 1 mm gaskets and three compressible gaskets to reach a stack thickness of 3200 μm (measured from the neutron images). The reactor design was slightly modified, without impacting the transport phenomena within the cell, by grooving the current collectors and the gaskets around the active area (Fig. 1c) to minimize the attenuation from cell parts with the neutron beam and thereby enhancing the sensitivity. After assembly, the cells were tightened with a torque-controlled wrench to 2 N m and the cell was mounted 1 – 3 mm in front of the neutron detector on a robotized platform. Peristaltic pumps (Cole-Parmer) were used to pump the electrolyte to the cells with rubber tubes (Masterflex LS-14 tubing) connected to two separate 20 mL electrolyte tanks. No inert atmosphere was used in the beamlines, the tanks were sealed with a rubber septum, limiting the oxygen availability in the system even though the presence of oxygen was not detrimental to the electrolytes employed[61]. Two flow rates were analyzed, 15.1 mL min$^{-1}$ and 5.6 mL min$^{-1}$ for the NEURA experiments and 21.1 mL min$^{-1}$ and 6.7 mL min$^{-1}$ for the ICON experiments, corresponding to superficial velocities within the flow field channels of 7.19 cm s$^{-1}$ and 2.67 cm s$^{-1}$, and 10.0 cm s$^{-1}$ and 3.19 cm s$^{-1}$, respectively, as calculated using Eq. (2).

$$u = \frac{Q}{N_{ch} W_{ch} D_{ch}} \quad (2)$$

Where $u$ is the fluid velocity [m s$^{-1}$], $Q$ is the electrolyte flow rate [m$^3$ s$^{-1}$], $N_{ch}$ is the number of inlet channels in the flow field configuration (7 channels) [-], $W_{ch}$ is the width of the channel ($1 \times 10^{-3}$ m) [m] and $D_{ch}$ is the depth of the channel ($5 \times 10^{-4}$ m) [m].

## Electrochemical protocols

Electrochemical measurements were conducted with a Biologic VSP-3e potentiostat. For the experiments in the NEUTRA beamline, the OCV of the cells was measured for 20 minutes (for the experiments with the KPF$_6$ supporting salt) or 1 hours (for the experiments with the BF$_4^-$ supporting ion) at 15.1 mL min$^{-1}$ after the cell was filled with the electrolyte to monitor the diffusion of species between the WE and CE. Thereafter, the cell was successively held at −0.3 V, +0.3 V, −0.6 V and +0.6 V for 20 min at each potential step. Then the flow rate was decreased to 5.6 mL min$^{-1}$ and the same potential protocol was applied. The entire electrochemical protocol was ca. 220 min and neutron radiographs were collected during the entire duration of the experiment. All measurements were performed in the same cell by filling and emptying the cell with the different solutions with a rinsing step in between with 0.5 M TEMPO dissolved in CD$_3$CN.

For the experiments at the ICON beamline, the OCV of the cells was measured for 10 min at 21.1 mL min$^{-1}$ after which the cell was successively held at −0.6 V and +0.6 V for 1 hour at each potential step. Then the flow rate was decreased to 1.7 mL min$^{-1}$ and the same

potential protocol was applied. Then the flow rate was increased to 6.7 mL min$^{-1}$ and the cell was successively held at −0.6 V, +0.6 V, −0.3 V and +0.3 V for 1 hour at each potential step. Finally, the flow-rate was switched to 96 mL min$^{-1}$ and the cell was held at −0.6 V and +0.6 V for 1 hour at each potential step. The entire electrochemical protocol took around 720 min to complete. For this study, only two flowrates (21.1 mL min$^{-1}$ and 6.7 mL min$^{-1}$) were analysed in detail. The electrolyte was not refreshed during the entire experiment.

## Neutron radiography

Neutron radiography experiments were performed at the NEUTRA thermal neutron and ICON cold neutron imaging beamlines at the Spallation Neutron Source (SINQ) facility of the Paul Scherrer Institute, Switzerland. In the SINQ facility, the neutrons were ejected from a lead spallation target that was hit with a proton beam at 590 MeV energy with 1.5 mA proton current. The ejected neutrons were moderated by heavy water and reached thermal velocities (with a mean energy of 25 meV)[62]. For the NEUTRA beamline, attenuated neutrons were captured by a scintillator screen (10 μm thick, Gd$_2$O$_2$S:Tb) and converted to visible light, which is subsequently captured by the charge-coupled device camera detector at an exposure time of 30 s. A tilted detector setup was used at the NEUTRA beamline that enables stretching in the horizontal transverse direction (with respect to the beam trajectory), see Fig. 1a, meaning that the membrane-electrode assembly can be imaged with higher spatial resolution[63]. The resulting pixel size in the direction across the membrane was 6 μm and the effective resolution, taking into account the blurring intrinsic to the detector and due to the beam divergence, was approximately 20 μm.

For the ICON beamline, neutrons were further moderated with a liquid deuterium (D$_2$) tank held at 25 K, decreasing the velocity of neutrons to the cold spectrum (mean energy of 8.53 meV)[64]. Similarly, the neutron beam passes through a series of collimators, beam limiters and shutters but before interacting with the sample the neutron beam passes through a mechanical chopper allowing ToF-NI. The chopper creates a pulsed neutron beam, whereas the travelled length of the neutrons through the flight tubes allows dispersion of the pulse based on the velocity of the constituent neutrons. The chopper rotated at a speed of 22 Hz with 4 regularly spaced openings, resulting in a pulse repetition frequency of 88 Hz. The angle of each opening was 18°, resulting in a 20% duty cycle. The path length between the chopper and the detector was 5.5 m. Finally, the neutrons were detected (exposure time 120 s per acquisition) at a microchannel plate detector having a fixed pixel size of 55 μm, an effective resolution of approximately 150 μm and a field of view of 28 × 28 mm$^2$ (512 × 512 pixel$^2$ images). In this way, the ToF-NI spectral technique in the ICON beamline allows spectral imaging of the sample, adding a new mode of contrast to conventional neutron radiography[50,65]. Per single ToF cycle of 11.36 ms (88 Hz of chopper disk rotation), 109 raw transmission images were taken. In total, 10560 cycles were completed over a total acquisition time of 2 min from high to low neutron energies, and the resulting images were binned over 109 images. Each of the recorded images represents the sum of the corresponding 10560 images acquired at one point of the cycle. Because of the relatively broad width of the neutron pulse (2.73 ms), each frame represents a blend of several neutron energies. This is not detrimental to the distinction of hydrogen and boron atoms, as the variation of neutron attenuation as a function of energy does not exhibit any stark feature. Frames were averaged from frames 16–30 to construct high energy (HE) images and from frames 31–50 to construct low energy (LE) images. The HE neutron energy ranges from 77.3–17.0 meV with a mean at 28.3 meV, and the LE neutron energy ranges from 16.0–5.7 meV with a mean at 8.5 meV. Since the image acquisition time is short compared to the experimental time (360 min), it can be assumed that HE and LE images are

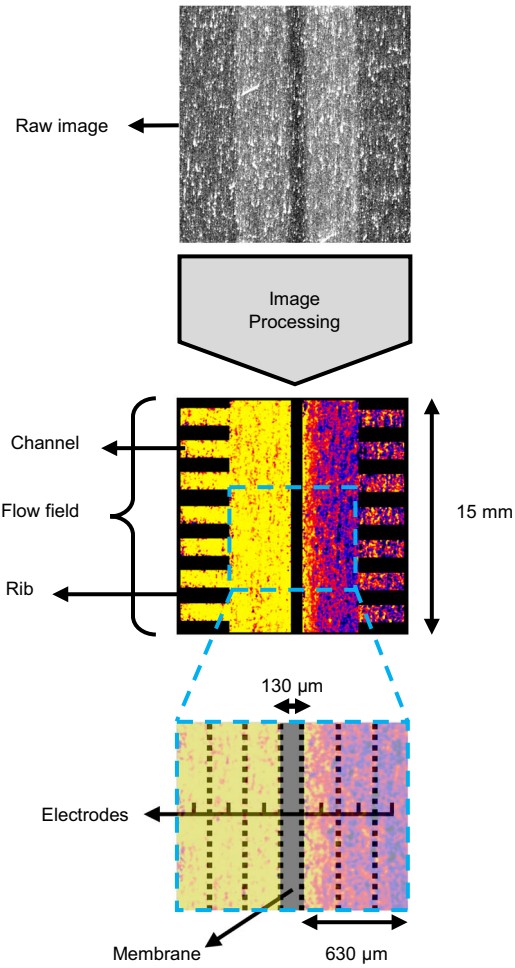

**Fig. 7 | Schematic representation of the image processing for the NEUTRA beamline experiments including the raw and final image.** The final image shows the flow fields, electrodes and membrane together with their dimensions.

taken at the same cell conditions. It was not possible to utilize the lower energy neutron spectra (frames 50–109) due to significant beam scattering and diffraction caused by the compression gaskets at their interface.

## Image processing

Image processing by mathematical calculations was performed to extract the sample information as described in the image processing pipeline (Fig. 7). Neutron radiography was performed in a subtractive manner, meaning that the transmission data from the samples were corrected for undesirable contributions, such as scattering and absorption of the cell components, detector background effects and beam instabilities. The image processing sequence for NEUTRA is shown in detail in Figure S9 and a description of the individual steps is given below:

1.  Dark current correction: Dark current images were taken with closed beam shutters and optical shutters of the camera to correct for the electronic bias within the camera circuitry.
2.  White spot filtering: White spots, resulting from gamma rays or other type of radiation hitting the camera chip, were filtered from the images by an outlier filtering approach, where pixels that deviate largely from their surroundings are replaced by a median value of these surroundings.
3.  Gaussian filtering: This filtering step reduces the statistical noise that is caused by high frequency photons hitting the camera.

4.  Open beam correction: Open beam images were taken without the electrochemical cell in the beamline to account for the spatial variation in beam intensity by dividing the images by the averaged open beam image.
5.  Registration: The registration step accounts for the physical movement of the cell during the experiments due to thermal dilation or relaxation of the cell body. All images were registered to a "reference" (the electrochemical cell filled with a 0.2 M $KPF_6$ in $CD_3CN$ solution) by applying a correlation algorithm on certain cell regions to find the optimum geometrical transformation.
6.  Intensity correction: This step corrects for beam intensity variations throughout the experiment duration by defining a non-changing area in the cell that corresponds to an area with no known changes (e.g., a part in the cell body). Gain and offset were applied to the images to match the intensity within the non-changing area region to that of the reference image.
7.  Scattered background correction: To account for the scattered background resulting from neutrons that are scattered by the cell and the detector, a scattered background image was obtained by interpolating the intensity values between so-called "black bodies". These "black bodies" are boron rods located in a grid fashion on a steel plate and are placed in front of the sample. The black body images were processed with all the processing steps described above, but the averaging on the images was performed before the registration and intensity correction steps to reduce the amount of noise. The final scattered background image was subtracted from the images to account for the scattered background.
8.  Referencing: In this step, the images were divided pixel-wise by the reference image that underwent all the processing steps above except registration and intensity correction, as these steps were based on the position and average intensity of the non-changing area in the reference image.
9.  Positioning: Images were rotated so that the flow channels were vertically aligned.
10. Cell mask: A cell mask was used to remove all the other parts of the image to obtain an image with only the region of interest, i.e., the membrane, electrode and flow channels.
11. Concentration calculation: The final transmission values obtained per-pixel were correlated to the species concentration via the Lambert-Beer law (Eq. (1)) as described in the Calibration experiments section, where the microscopic cross-sections were obtained from the cuvette measurements and with $\delta_{cell}$ the electrolyte thickness within the cell [m] which is a function of the cell geometry, electrode porosity and compression according to:

$$\delta_{cell} = L_e \varepsilon_e \qquad (3)$$

where $L_e$ is the electrode width ($1.7 \times 10^{-2}$ m) [m] and $\varepsilon_e$ the electrode porosity at the applied compression (which is 85.3% for three stacked SGL 39AA paper electrodes) [–]. The total concentrations extracted from neutron imaging were corrected using reference values from the supporting salt solutions, which have a higher solvent molarity compared to the solution containing the redox-active species. This effect was not accounted for, resulting in a slight underestimation of the concentration maps.

Due to the nature of the microchannel plate detector and the ToF-NI method used at the ICON beamline, the image processing sequence is slightly different than for NEUTRA. Overlap correction was performed on all images before the image processing steps due to the characteristics of the microchannel plate detector[66]. The image processing sequence to obtain transmission images is listed below:

1.  Outlier removal: Dead pixels (or zero pixels) were removed from the images by averaging pixels around them and setting the new value to it.

2. Scrubbing correction: To avoid bias due to the detector efficiency over time, open beam images were used to correct for any change of contrast not related to the experiment but to the detector. The function interpolates between open beam images and sets a weight to correct them.

3. Scattered background correction: Although microchannel plate detectors have less contribution of scattered background due to the transmission of neutrons parallel to the beam-axis, we still performed scattered background correction to improve accuracy[67]. For ICON, the black bodies were strips of Boral (2.5 mm width) in a steel plate.

4. Registration

5. Binning: Selection and merging of the HE and LE images based on the energy-dependent calibration curves of the neutron attenuation of the different species of interest and accounting for the edge scattering effects.

6. Intensity correction

7. Referencing

8. Positioning

9. Cell mask

10. Concentration calculation: The concentration of each species was obtained by solving a system of equations for the transmission of each region of interest (see Fig. 5b), where the LE and HE microscopic cross-sections of the species can be correlated to the concentration distribution of the *operando* images via the following operation

$$\begin{bmatrix} -\ln(T_{LE}) \\ -\ln(T_{HE}) \end{bmatrix} = \delta_{cell} \cdot \begin{bmatrix} \sigma_{BF4,LE} & \sigma_{TEMPO,LE} \\ \sigma_{BF4,HE} & \sigma_{TEMPO,HE} \end{bmatrix} \cdot \begin{bmatrix} C_{BF4} \\ C_{TEMPO} \end{bmatrix} \cdot N_A \quad (4)$$

where $T_{LE}$ and $T_{HE}$ are the low energy and high energy transmission values per-pixel of the *operando* experiment images [−], $\sigma_{i,LE}$ and $\sigma_{i,HE}$ are the low energy and high energy microscopic cross-sections [m$^2$], and $C_i$ is the pixel-wise concentrations of the species $i$ [mol m$^{-3}$], which are TEMPO or BF$_4^-$. The electrolyte thickness, $\delta_{cell}$, is defined according to Eq. (3), which for the felt electrode used in the ICON experiments is around 1.62 cm.

## Data availability

The data presented in this study can be provided by the corresponding author upon request. Source data files are provided with this paper. The data in the figures are provided as Source Data Files and can be obtained in the ChemRxiv (https://chemrxiv.org/engage/chemrxiv/article-details/64e77958dd1a73847f6d8c15). Supplementary data can be found in the Supporting Information and two videos are available online.

## Code availability

The code used in this study can be provided by the corresponding author upon request.

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

## Acknowledgements

A.F.C. gratefully acknowledges funding by the European Union (ERC, FAIR-RFB, ERC-2021-STG 101042844). Views and opinions expressed are however those of the author(s) only and do not necessarily reflect those of the European Union or the European Research Council. Neither the European Union nor the granting authority can be held responsible for them. A.F.C. gratefully acknowledges the Dutch Research Council (NWO) for financial support through the Talent Research Program Veni (17324) and the NWO Science XS grant (OCENW.XS4.295). E.B.B.

gratefully acknowledges funding from EIRES. The results of this project are based on experiments performed at the Swiss Spallation Neutron Source, SINQ, Paul Scherrer Institute (Switzerland), with proposal numbers 20202189 and 20202333. This work was partially supported by the Joint Center for Energy Storage Research (JCESR), an Energy Innovation Hub funded by the United States Department of Energy. K.V.G. acknowledges additional funding from the National Science Foundation Graduate Research Fellowship under grant no. 1122374. Any opinion, findings and conclusions or recommendations expressed in this material are those of the authors and do not necessarily reflect the views of the National Science Foundation. A.F.C. acknowledges financial support from the Swiss National Science Foundation (P2EZP2_172183) during his postdoctoral research at MIT. The authors are thankful to Adrian Mularczyk (Eindhoven University of Technology), Katelyn Ripley and Trent Weiss (both from Massachusetts Institute of Technology) for their feedback on the manuscript.

## Author contributions

R.R.J., M.v.d.H. and E.B.B. contributed equally to this work. R.R.J. contributed to the conceptualization, methodology, formal analysis, investigation, data curation, visualization, writing-original draft and writing-review and editing. M.v.d.H. contributed to the conceptualization, methodology, formal analysis, investigation, data curation, visualization, image processing, writing-original draft and writing-review and editing. E.B.B. contributed to the conceptualization, methodology, formal analysis, investigation, data curation, visualization, writing-original draft and writing-review and editing. E.R.C.R. contributed to the methodology, formal analysis, investigation, data curation, image processing and writing-review and editing. K.V.G. and J.A.K. contributed to the conceptualization, methodology and writing-review and editing. V.M.P. contributed to the visualization and writing-review and editing. F.R.B. contributed to funding, resources, writing-review and editing. K.N. contributed to the funding, resources, writing-review and editing and supervision. P.B. contributed to the conceptualization, methodology, formal analysis, investigation, data curation, image processing and writing-review and editing. Finally, A.F.C. contributed to the conceptualization, methodology, formal analysis, investigation, data curation, funding, resources, writing-original draft, writing-review and editing, project administration and supervision.

## Competing interests

The authors declare no competing interests.

## Additional information

[1]Electrochemical Materials and Systems, Department of Chemical Engineering and Chemistry, Eindhoven University of Technology, P.O. Box 513, 5600 MB Eindhoven, The Netherlands. [2]DIFFER - Dutch Institute for Fundamental Energy Research, P.O. Box 6336, 5600 HH5612 Eindhoven, The Netherlands. [3]Eindhoven Institute for Renewable Energy Systems, Eindhoven University of Technology, P.O. Box 513, 5600 MB Eindhoven, The Netherlands. [4]Electrochemistry Laboratory, Paul Scherrer Institut, Forschungsstrasse 111, CH-5232 Villigen PSI, Switzerland. [5]Joint Center for Energy Storage Research, Massachusetts Institute of Technology, Cambridge, MA 02139, USA. [6]Department of Chemical Engineering, Massachusetts Institute of Technology, Cambridge, MA 02139, USA. [7]Department of Thermal and Fluids Engineering, Universidad Carlos III de Madrid, 28911 Leganes, Spain. [8]Membrane Materials and Processes, Department of Chemical Engineering and Chemistry, Eindhoven University of Technology, P.O. Box 513, 5600 MB Eindhoven, The Netherlands. [9]Laboratory for Neutron Scattering and Imaging, Paul Scherrer Institut, Forschungsstrasse 111, CH-5232 Villigen PSI, Switzerland. [10]These authors contributed equally: Rémy Richard Jacquemond, Maxime van der Heijden, Emre Burak Boz. ✉e-mail: a.forner.cuenca@tue.nl

