## [Peer Review File · Nature Communications]

REVIEWER COMMENTS

Reviewer #1 (Remarks to the Author):

Novelty

The use of neutron imaging to visualize concentration profiles in operando flow batteries has not been demonstrated before. Neutron imaging has been used to study other phenomena such as bubbling in fuel cells. Concentration profiles have not been studied in a similar way. It is a novel method of visualizing the phenomena taking place inside flow batteries.

Suggested improvements

Even though the proposed method allows unprecedented visualization of the concentration profiles within the flow batteries it would be interesting to see what practical information could be extracted. In this direction, could the authors answer the following questions?

-How could the insight into the transport processes translate into practical information on flow batteries which could guide the researchers in the field?

-Are there any parameters that relate to transport properties of the species or performance which could be extracted from the neutron images?

Summary

The work is quite novel and very well presented; however, it seems that it lacks practical significance. Besides insight, the authors do not mention any practical implications of their method.

Due to the lack of practical significance, I would not recommend publication in Nature Communications. That said, it is suitable for publication in a more specialized journal.

Reviewer #2 (Remarks to the Author):

In this manuscript, the authors report the application of neutron radiography in redox flow cell. The concentration distributions of redox active species and supporting salts were visualized in operando redox flow cells. This technique can also be extended to other systems involving reactive transport phenomena. It's an interesting and important work. I'd like to recommend it for publication after minor revisions.

1.The correlation between the concentration distributions of redox active species and supporting salts, and the macroscopic performance needs to be further strengthened.

2.The references need to be updated.

Reviewer #3 (Remarks to the Author):

The authors developed a novel neutron radiography approach to quantify the concentration of species in solution during electrochemical operation in redox flow cells. They demonstrated for the first time the use of neutron radiography to image concentrations of redox active species and supporting salts in operando electrochemical flow cells. This is an excellent contribution to the field and the reported method and results are important guides for future development of advanced characterization for flow batteries. The schematics are excellent and very clear. I recommend to publish this with minor revision and here are my comments.

1. It was mentioned “Using a low attenuating supporting salt (KPF6), we selectively image the motion of redox active species...” this means that salt/active materials combination is important to ensure high selectivity in imaging? it would be helpful to discuss what are the materials constrains for salt, and active materials to be examined using this method, so readers can understand how to apply this method in their system.

2. I am wondering would the materials selection for gasket/current collector affect the measurement?

3. I am particularly interested in understanding if this technique can be applied in polysulfide based flow batteries (<https://doi.org/10.1038/s41560-023-01370-0>, <https://doi.org/10.1038/s41560-021-00804-x>) and heteropoly acid based flow batteries (<https://doi.org/10.1038/s41560-022-01011-y>), For polysulfide RFBs, for instance, it would be extremely interesting to examine the concentration profile near the electrode/membrane interfaces to understand the effect of membrane and molecular catalyst on the reaction. It would be helpful to elaborate in the perspective if possible.

4. Is the cell entirely sealed during the examination? As some organic molecules are sensitive to air/oxygen.

5. A minor point, the reference may have some issues showing “Error! Reference source not found.”

Authors' responses to the Reviewers' comments

In the following, we address point-by-point the reviewers' comments. We use the following style:

- Reviewer comments are listed in “***bold-Italics***”
- Our responses follow in “normal text.”
- Excerpts from the original manuscript submission are presented in *italicized and blue font*.
- Changes in the revised manuscript text are shown in *highlighted and italicized in blue font*.
- Furthermore, we performed a couple of thorough revisions of our manuscript (*highlighted*) to identify and correct several typos and to improve the clarity of the text where possible.

Reviewer #1: The use of neutron imaging to visualize concentration profiles in operando flow batteries has not been demonstrated before. Neutron imaging has been used to study other phenomena such as bubbling in fuel cells. Concentration profiles have not been studied in a similar way. It is a novel method of visualizing the phenomena taking place inside flow batteries.

Even though the proposed method allows unprecedented visualization of the concentration profiles within the flow batteries it would be interesting to see what practical information could be extracted. In this direction, could the authors answer the following questions?

Author response: We thank the reviewer for their positive and constructive feedback which has improved the quality of the revised manuscript. Below, we address in detail the comments of the reviewer with a particular focus on the practical information that can be extracted with the method.

(Comment #1.1): How could the insight into the transport processes translate into practical information on flow batteries which could guide the researchers in the field?

Author response: We appreciate the opportunity to elaborate on how the presented method can be leveraged to extract meaningful practical information to guide flow battery researchers and beyond. In this first publication, we have focused on describing the method in detail and applying the method to one specific type of redox flow cell reactor architecture (e.g., non-aqueous redox chemistry) with the goal of visualizing concentration distributions during electrochemical operation. With this approach, we hoped to motivate the use and extension of this technique by other groups, while also demonstrating the broad impact of the technique which can be applied to many (electro)chemical engineering problems driven by mass- and/or convective transport.

With this in mind, and following the comment of the reviewer, we here expand on the practical impact of the methodology. We illustrate practical insights by using four examples of neutron imaging experiments from our group, including mostly unpublished data.

Example 1: Influence of membrane type on reactive transport phenomena

In this study we have limited our scope to anion exchange membranes to increase focus and impact. In our ongoing research, we used the developed approach to investigate the impact of various membrane types (e.g., porous separators, anion exchange membranes, cation exchange

membranes) on the concentration distribution in the electrochemical system. **Figure R1** presents a comparison of concentration distributions for TEMPO/TEMPO⁺ redox species and BF₄⁻ supporting salt in the reactor areas of two RFBs, one operated with a porous separator (**a**) and the other with an anion-exchange membrane (**b**). From a chemical perspective, porous separators are expected to have no charge selectivity, while anion-exchange membranes should predominantly allow negatively charged species to pass. Membrane properties are generally determined via H-type cells where diffusion and convection influence the species transport. Conventional methods measure average concentrations in a given volume, but cannot resolve the concentration distributions that arise within the reactor due to migration effects. Neutron radiography enables visualization of these membrane selectivity effects. In **Figure R1a**, the concentration profiles of TEMPO/TEMPO⁺ and BF₄⁻ exhibit mirrored fluctuations in response to applied voltage. This aligns with expectations, as the porous separator lacks charge selectivity, allowing both TEMPO⁺ and BF₄⁻ to function as charge carriers. In **Figure R1b**, only the BF₄⁻ concentration fluctuates with applied potential due to the Donnan exclusion of TEMPO⁺ molecules at the membrane-electrolyte interface. This example illustrates how neutron radiography facilitates the visualization of membrane effects on transport phenomena at the reactor level. We anticipate that this methodology will contribute to the design of novel membrane materials by aiding researchers in comprehending the impact of materials on electrochemical properties through the unravelling of local concentration effects.

Figure R1 – Energy selective imaging enables the deconvolution of TEMPO species and BF₄⁻ concentrations. (a) Concentration profiles of TEMPO species and BF₄⁻ in the WE and in the CE compartment. The added images correspond to: OCV (64 images averaged), -0.6 V, +0.6 V at 21.1 mL min⁻¹ and -0.6 V, +0.6 V (8 images averaged) at 6.7 mL min⁻¹ (from left to right) (b) Energy selective imaging of a NAqRFB with FAB-PK-130 anion-exchange membrane. The NAqRFB consisted of initially 0.5 M TEMPO in the CE and 0.5 M TEMPO⁺BF₄⁻ the WE. The snapshots correspond to a full period average for the OCV and 4 images average for the different potential steps.

Practical information: We can correlate the membrane properties (i.e., ion-selectivity) with the mass transport through the electrochemical reactor using neutron radiography. The increased species flux between compartments through migration, coupled with visualization of the local concentration with neutron radiography allows us to investigate the concentration polarization within an electrochemical cell. We can use this approach to study the effect of individual reactor components (membranes, flow fields, electrodes) and operating parameters (flow rate, voltage, current) on the mass transport through the reactor, additionally coupled to the macroscopic performance of the flow cell. Such studies aid in the understanding of the structure-performance relationships of the reactor components which are required for the selection or design of materials for specific reactor designs and operating conditions.

Example 2: Visualization of wetting behaviour of porous electrodes

Instead of visualizing species concentrations, we can also use neutron imaging to visualize pore utilization by a liquid electrolyte. Such approach enables visualization of porous electrode wetting, which can be correlated to electrochemical parameters of interest, such as the electrochemically active surface area (ECSA). In a previous work, we reported the use of neutron radiography to understand the wetting dynamics of porous electrodes that were electrografted with a hydrophilic molecule (taurine) to enhance their wettability and promote better reactor utilization. **Figure R2(a)** is a representation of the experimental setup used for the measurement and is similar to the one described in the present manuscript. The left electrode was a carbon cloth not subjected to any treatment as opposed to the right electrode which had been electrografted with taurine. The electrode water thickness was extracted from neutron images to calculate the electrode saturation as shown in **Figure R2(b)**. The wettability of the commercial electrode remained poor unless high electrolyte velocities were used. On the contrary, the taurine-grafted electrode almost reached full electrode saturation even at the lowest electrolyte velocity, indicating superior electrode wettability.

Figure R2 – Neutron radiography experiment to visualize electrode wetting operando in a flow cell. (a) Scheme of the in-plane neutron imaging where untreated and taurine-treated ($\times 50$) electrodes, separated by a dense PTFE film, are enclosed by flow-by flow fields and plastic housing in a flow cell setup. (b) Water thickness and saturation of untreated and taurine-treated ($\times 50$) cloth electrodes over time where the electrolyte velocity in flow channels is increasing in stages. A snapshot of each velocity zone is given on top of the graph where yellow depicts dry areas and blue depicts the electrolyte.[1]

Practical information: We can correlate electrode saturation at the local scale with important macroscopic metrics (e.g., electrochemical active surface area, effective ionic resistance). We can use this information to design better performing flow field geometries, porous electrode microstructures and surface properties, as well as flow conditions. We envision that in-plane neutron radiography during electrochemical operation will prove to be useful for electrochemical devices where multiple phases coexist within the reactor.

Example 3: Iron plating distribution with polarized neutrons

Hybrid flow batteries are one example where multiphase transformations complicate the device operation. We recently investigated the hybrid all-iron RFB system through imaging with polarized neutrons. By placing a spin filter in the beam path, we controlled the magnetic spin of neutrons and were able to image samples having ferromagnetic changes during electrochemical operation. In **Figure R3** we show the three first charging cycles of a full iron redox flow battery observed with polarized neutrons. In the first image of the second cycle, the reactor area has been highlighted, the left rectangle and right rectangles are separated by a membrane and correspond to the two-electrode region. As it can be seen from the images, the iron plating (dark regions) commences near the membrane and spreads towards the flow field of the cathodic compartment. In the first cycle only a fraction of the electrode area is plated with iron as most of the current is directed towards hydrogen evolution, thus explaining the low faradaic efficiency in cycle 1. The subsequent cycles show a larger fraction of solid iron in the negative electrode as the electrolyte pH increases and iron plating becomes more efficient.

Figure R3 – Polarized neutron images during the first 3 cycles of a full-iron RFB cycling. The electrodes were two thermally treated AvCarb 2.8 mm felts and the membrane was a Nafion 212. Cycles were performed at 100 mA cm^{-2} using flow-through flow fields.

Practical information: We can track the iron plating reactions in the flow cell over multiple plating and stripping cycles. Increased resistance due to membrane fouling is a reported phenomenon for hybrid all-iron RFBs. [2] Thus, one example use of neutron radiography would be to investigate how iron plating can be limited on the membrane. [3] We can use this approach to investigate the plating and stripping dynamics (i.e., in time and across the electrode volume) under various operating conditions and cell configurations (e.g., using various electrodes, spacers, and flow fields). This could also be applied to imaging industrially-relevant plating processes.

Example 4: *Detection of secondary phenomena such as salt precipitation*

Finally, secondary phenomena, including salt precipitation and underutilization of the flow field channels, can be tracked with neutron radiography. In **Figure R4**, we observed salt precipitation in the porous electrode and flow field channels as a result of not rinsing the cells between the KPF₆ supporting salt and BF₄⁻ counter ion experiments. We observed that the BF₄⁻ ions precipitate when in contact with K⁺ ions due to the low solubility of KBF₄ in acetonitrile, resulting in the dark spots visualized in **Figure R4** [4]. These observations were useful for the experimental design of the article under review, and we added a rinsing step between experiments (with a 0.5 M TEMPO solution) to prevent salt precipitation.

Figure R4 – Concentration snapshots of TEMPO species and BF₄⁻ in the WE and in the CE compartment under various applied potentials to show the formation of salt precipitates. Where the counter electrode corresponds to the 0.5 M TEMPO (left half of the graphs) and the working electrode to the 0.5 M TEMPO⁺BF₄⁻ (right half of the graphs). The images correspond to: OCV, +0.3 V, -0.3 V, +0.6 V, -0.6 V at 21.1 mL min⁻¹.

Practical information: Using neutron radiography, secondary phenomena in the flow cell can be observed, such as salt precipitation. *These observations are crucial in the understanding and improvement of RFBs operating close to their solubility limits (vanadium) or batteries where insoluble multi-species complexes can be generated (polysulfide).*

To summarize, the present manuscript focuses on describing the foundations of the neutron imaging methodology applied to the field of RFBs. This work has undoubtable novelty, and the developed practical examples (although outside the scope of the current work) will certainly pave the way to new technological development in system engineering and novel materials applied to RFBs.

To improve the quality of the manuscript and its impact on the RFB community we added a section concerning the practical relevance of the presented neutron radiography approach (page 24):

Practical application of the neutron imaging method

In this work, we have applied the new neutron imaging methodology to study non-aqueous redox flow batteries using multicomponent electrolytes. Beyond this specific application, we anticipate numerous applications of neutron imaging for quantifying concentration distributions in electrochemical cells and beyond. First, the resulting concentration maps can be used as experimental data to validate computational models that describe reactive mass transport. Here we used an electrolyte composed of a solvent, supporting electrolyte, and two redox active species - hence a complex, multicomponent system close to practical devices; but model experiments can be performed (e.g. with one or two analytes) to systematically deconvolute mass transport modes (e.g. diffusion, convection and migration) and their associated transport rates. Deeper fundamental understanding of reactive mass transport in electrochemical reactors and through membranes will assist designing advanced electrochemical cells. Second, the methodology enables identification of local maldistributions in concentration, which can assist in designing better flow field geometries and electrodes, as well membrane crossover, Donnan exclusion and salt precipitation within the electrochemical cell, which are deleterious to performance and lifetime in several flow battery chemistries (e.g. non-aqueous, all-vanadium and all-iron). Third, we anticipate that the method - and adaptations on the detection physics - will be instrumental in advancing hybrid redox flow batteries (e.g. all-iron, zinc-bromine), where there are phase change reactions (e.g. plating and stripping or hydrogen evolution) fundamentally limiting the performance of the system. Fourth, we anticipate that the technique can be applied to technical systems such as electrochemical stacks, where traditional neutron imaging was instrumental in advance fuel cell stacks through visualization of water distributions. Fifth, beyond redox flow batteries, the method can be applied to other (electro)chemical reactors where concentration profiles determines performance such as electrochemical separations, flow chemistry, and chemical reactor design.

Finally, the use of molecularly engineered redox molecules acting as imaging probes might enable simultaneous visualization of concentration of multiple components (>2) when combined with energy-selective neutron imaging. Although the technique is still in its early stages, it displays considerable potential. We anticipate that ongoing advancements in neutron detectors and choppers will enable more sophisticated analyses of complex multicomponent systems using ToF-NI, offering enhanced spatial, temporal and energy resolution.

In addition, the following changes were made to the manuscript:

Main manuscript page 15:

The experiment begins with an OCV step where no current is drawn from the cell. The brighter colour of the CE side in the OCV radiograph represents a higher concentration compared to the WE side, as expected by the concentrations of the electrolyte fed (0.5 M and 0.2 M TEMPO/TEMPO⁺). An advantage of neutron radiography is that electrolyte wetting of the porous electrodes can be visualized because of the low attenuation of gasses, which will appear as dark spots (i.e., lower concentration) in the radiographs [64]. During the OCV period, the concentration on both sides does not show dark regions (Figure 4c-d), suggesting full wetting, at least to the spatial resolution of the measurement.

Main manuscript page 18:

At the WE, we find higher concentrations in the areas near the flow field inlets in comparison with the area under the ribs, showing an advantage of two-dimensional concentration maps obtained using neutron radiography.

(Comment #1.2): Are there any parameters that relate to transport properties of the species or performance which could be extracted from the neutron images?

Author response: We thank the reviewer for the question. In brief, it is possible to extract mass transport properties from the neutron radiographs. The 2D concentration maps obtained from the neutron radiographs can be seen as a matrix of concentrations values such as $C_i(x,y,t)$. At each location in the reactor (x,y) we obtain one average concentration value C_i at a particular time (t). Although beyond the scope of this work, one can use these concentration maps as input for models that describe momentum, mass and charge transfer. By designing model experiments (i.e. with a single component or two components with distinct attenuations), we can isolate and quantify specific transport mechanisms (e.g., diffusion, migration, reaction) from which transport parameters can be obtained (e.g., diffusive/migration/convective rates).

However, in this paper, we elected to demonstrate the novel neutron imaging methodology applied to a practical electrochemical cell (including a three-component electrolyte which represents practical operation). With this approach, we are able to demonstrate the application to real systems, but compromise on the ability to extract individual transport parameters, which should be the focus of a future dedicated study on mass transport.

As visualized in **Figure R5**, various transport processes simultaneously take place in operating flow cells. The concentration distributions obtained in this work are thus a combination of these processes and can therefore not directly be correlated to individual transport properties/parameters, as for example diffusive rates. Nevertheless, individual transport contributors, such as those contributing to the species transport, can be investigated by **clever design of experiments**. For example, for a multicomponent system investigated with a white neutron beam (NEUTRA), diffusion rates of neutron active compounds could be determined by injecting a known quantity of analyte in the electrolyte solvent and follow the concentration evolution in the reactor area over time without applied potentials. When this experiment is followed by experiments similar to the ones performed in **Figure 4** of the main manuscript under convective operation with and without applied potentials, convection and migration rates can be obtained by comparing the neutron images at the same conditions (e.g., electrolyte concentration and fluid dynamics) with and without applied electric field. Another approach would be to utilize different reactor configurations, for example using various types of (charged) membranes, as presented in **Figure R1**, which enable only the passing of selected ions, allowing the studying of diffusion and migration between the half-cells. Another approach would be the coupling of the neutron radiography concentration profiles to **computational models**, for example to 2/3-dimensional macroscopic continuum models that can resolve the individual transport rates (e.g., using the Nernst-Planck equation). Although possible, this approach is tedious and time consuming because of the model validation (including the extraction of experimental parameters required for the model) and the poor selectivity of the neutron radiography with white beam towards the different components of the system. On the contrary, neutron radiography experiments could also be used to validate computational models as the concentration profiles obtained from the experiments can be compared and fitted to the computational modelling results to allow simulation under realistic conditions.

Figure R5 – Schematic representation of the transport mechanisms present in a flow cell, where the colors depict the distinct transport mechanisms (momentum, species, charge, and heat transport). The figure was obtained from van der Heijden *et al.* [5].

Furthermore, neutron radiography is a powerful tool to obtain performance indicators of the reactor. Several were already mentioned in **Comment #1.1**, including the visualization of secondary phenomena in the reactor such as salt precipitation and uncomplete filling of flow field channels, electrolyte infiltration in the porous electrode, or metal plating throughout the porous electrode.

Additionally, constant advances in neutron imaging hardware and optical configurations help pushing the boundaries of what can be observed and extracted from neutron radiography experiments as highlighted in **Figure 7** of the main manuscript. By using a cold neutron spectrum coupled with the addition of a time-of-flight (ToF) tube in the optical hardware, energy selective neutron imaging became possible. With this imaging configuration one can selectively image the concentrations of individual components in the reactor area, thus transforming a complex multicomponent system to a sum of simple single-component systems. Finally, we want to put the emphasize on the novelty and the potential of the presented approach to assist the design of novel materials and reactor components in electrochemical technologies. The presented results of energy-selective neutron imaging were acquired on a cutting-edge hardware at the SINQ facilities and we envision that the ongoing research and development on neutron detectors with higher spatial resolution and choppers with faster duty cycles enables better neutron energy separation.

To improve the quality of the manuscript and its impact on the RFB community we added a section concerning the practical relevance of the presented neutron radiography approach (page 24) and the previous comment.

In addition, the following alterations were made to the document:

Main manuscript page 14:

The cell is discharged (negative potential applied at the WE) and charged (positive potential applied at the WE) alternately, such that the state-of-charge after each complete cycle does not significantly deviate from the initial condition and two voltage magnitudes were applied to understand their impact on the potential-driven transport processes (e.g., migration).

Main manuscript page 15:

Moreover, over the course of the OCV period, the concentration profile remained fairly constant which can be attributed to the low diffusion rate of TEMPO/TEMPO⁺ through the dense anion exchange membrane.

Main manuscript page 17:

This results in a stronger diffusive flux of TEMPO towards the WE side at positive potentials, supported by the bright concentration front in the corresponding radiographs, together with the steep concentration profiles near the membrane. Increasing the potential to +0.6 V amplifies this trend as more TEMPO⁺ is converted to TEMPO on the CE side, exacerbating the concentration gradient of TEMPO and extending the concentration front deeper within the WE (Figure 4d).

Main manuscript page 17:

To visualize such limiting phenomena, we utilized a flow-by flow field design that induces limited convection within the porous electrode. However, the intensification of the concentration fronts suggests that this flow field does have convective transport contributions in the electrode. Nevertheless, we anticipate that a convection-enhanced flow field (such as interdigitated or flow-through) would further increase species replenishment and reduce concentration gradients [65].

Main manuscript page 17:

This resonates with the subtle changes in the concentration profiles in Figure 4 as a function of the applied potential, hinting that the neutron-transparent PF₆⁻ is the main charge carrier in the system. A novel approach to isolate the concentration of species in solution with neutron imaging is discussed in the energy selective imaging section.

Main manuscript page 18:

When a positive potential is applied to the WE, the reverse reactions take place, and the resulting radiographs and the concentration profiles (Figure 5e) are mirrored compared to negative applied potentials. Because of the reactor architecture used in the NEUTRA experiments (stacked paper electrodes and a flow-by flow field) in combination with the low ionic conductivity of the electrolyte, there is only a small change in the cell capacity at negative potentials, consuming only ~8% of the total capacity (Figure S6), resulting in relatively low current densities. Therefore, at positive potentials, only a small amount of TEMPO is present in the WE compartment to be converted back to TEMPO⁺. As a result, large overpotentials are generated throughout the cell due to the low concentration of reactants to sustain the current. This explains the asymmetry in the current magnitudes when the polarity of the cell is reversed (i.e., +7 mA vs. -65 mA at 15.1 mL min⁻¹ and +/-0.3 V, Figure 5b), and the even lower capacity recovery (~1-2%) resulting in underutilized capacity over the duration of the experiment (Figure S6).

When comparing the experiments with counter-ion BF₄⁻ and supporting salt KPF₆, we can correlate the macroscopic performance with the concentration distributions through the reactor. For the KPF₆ experiments, all active species (i.e., TEMPO, TEMPO⁺, PF₆⁻) are present in both compartments. Therefore, the macroscopic performance, i.e., the current output, is symmetric when operating at negative and positive applied potentials (Figure 4b) as the to-be-reacted species are present without the requirement of species crossover under the evaluated conditions (as the capacity change is limited to ~20%). The symmetric current output

results in concentration profiles returning to the OCV profile when positive potentials are applied. Whereas for the BF_4^- experiment, the charged species are only present in one compartment (WE) initially and are required to cross the membrane to support the reactions, which is limited by the anion exchange membrane, resulting in asymmetric current magnitudes upon changing cell polarities (**Figure 5b**). The asymmetric current can be correlated to the concentration profiles as for positive applied potentials, the concentration does not fully return to the OCV profiles (**Figure S7**).

Main manuscript page 22:

These results corroborate the observations from the experiments in NEUTRA as only minor concentration fluctuations were observed with PF_6^- as supporting salt when an electric field is applied, whereas stark changes are detected when the supporting ion was changed to BF_4^- .

Summary: The work is quite novel and very well presented; however, it seems that it lacks practical significance. Besides insight, the authors do not mention any practical implications of their method. Due to the lack of practical significance, I would not recommend publication in Nature Communications. That said, it is suitable for publication in a more specialized journal.

Author response: We thank the reviewer for their constructive and insightful feedback. We agree with the reviewer that our first submission could have more strongly reflected on the practical significance of the work, and have made strong efforts in highlighting the practical significance in the revised version. Notably, we have added a new section describing practical applications of the method in the main manuscript, and have included an extensive set of practical guidelines (hardware materials, attenuation of many redox chemicals and supporting electrolytes) in the SI. We hope that these two additions will motivate other researchers to explore additional applications of the technique and will lower the barrier to use the technique thanks to the practical guidelines.

In summary, we strongly believe the method is poised to play a major role in developing practical applications such as redox flow batteries, electrochemical separations, flow chemical reactors and other applications in the chemical industry. We anticipate that quantification of species concentrations using neutrons in practical device architectures (even in full stacks) will be an instrumental technique in advancing technologies, akin to the advancement of fuel cell stack technology in the 2010s, which was largely supported by neutron imaging of water.

Reviewer #2: In this manuscript, the authors report the application of neutron radiography in redox flow cell. The concentration distributions of redox active species and supporting salts were visualized in operando redox flow cells. This technique can also be extended to other systems involving reactive transport phenomena. It's an interesting and important work. I'd like to recommend it for publication after minor revisions.

Author response: We thank the reviewer for their kind words and constructive feedback which has improved the quality of the revised manuscript.

(Comment #2.1): The correlation between the concentration distributions of redox active species and supporting salts, and the macroscopic performance needs to be further strengthened.

Author response: We thank the reviewer for the interest in the correlation between cell performance and neutron images. Following the recommendation by the reviewer, we have made an effort to strengthen the correlation, which we explain below and made additional changes throughout the manuscript.

Because we combined *operando* neutron radiography with *operando* electrochemical diagnostics, there is a direct correlation between the macroscopic performance and the concentration distributions within the flow cell. These can be observed when comparing for example the results of the NEUTRA experiments with and without supporting salt as a result of the chosen reactor architecture and electrolyte concentrations. As a result of the anion exchange membrane, only negatively charged and neutral species can cross the membrane, allowing the supporting ions to pass, but not the TEMPO⁺ species. In **Figure 4** in the manuscript, all active species (i.e., TEMPO, TEMPO⁺, PF₆⁻) are present in both compartments. Therefore, the macroscopic performance, i.e., the current output, is symmetric when operating at negative and positive applied potentials as the to-be-reacted species are present without the requirement of species crossover under the evaluated conditions (as the capacity change is limited to ~20%). Whereas in **Figure 5**, the charged species are only present in one compartment (WE) initially and are required to cross the membrane to support the reactions. However, the TEMPO⁺ active species are prohibited to cross the membrane as a result of their positive charge. Therefore, as only the fraction of TEMPO⁺ that was formed at the negative applied potentials (~8%) is present to react, large overpotentials are generated throughout the cell due to the low concentration of reactants to sustain the current, which explains the asymmetry in the current magnitudes when the polarity of the cell is reversed (i.e., +7 mA vs. -65 mA at 15.1 mL min⁻¹ and +/-0.3 V, **Figure 5b**).

Although beyond the scope of the current work, the correlation between the concentration distributions and macroscopic performance metrics becomes more apparent when comparing distinct reactor components. For example, when comparing different types of membranes in **Figure R1**, stark differences can be observed in the current output as a result of the restriction of certain ions to pass the ion-selective membranes. In another ongoing work we correlate the macroscopic performance to the concentration profiles for various electrodes and flow fields where we find that by tracking the cumulative TEMPO/TEMPO⁺ movement, a bimodal pore size distribution with large through-plane voids with a low tortuosity is beneficial for parallel flow field configurations compared to unimodal electrodes, as the large voids improve through-plane electrolyte transport, observed by less sloped concentration profiles over the electrode thickness, and warrant high current densities, observed from the current profiles over time. However, because of the resolution of the neutron radiography approach used, local concentration variations (<10 μm) cannot be resolved and would require local imaging

diagnostics (e.g., fluorescence/confocal microscopy [7]) which can subsequently be used to study microstructural effects (diffusion boundary layers, local reaction rates, local electrolyte velocities) in depth to explain the significant differences at the reactor level that cannot solely be explained by the concentration distributions obtained in this work.

The following additions were made in the main manuscript:

Main manuscript page 17:

In the first hour of the experiment, the system is kept at OCV conditions to track diffusional crossover through the membrane (Figure 5b). Although we track a change in OCV over time, indicating crossover and concentration equilibration, the small concentration variations in this short time period are not quantitatively captured by the radiographs (OCV radiograph in Figure 5c).

Main manuscript page 17:

This resonates with the subtle changes in the concentration profiles in Figure 4 as a function of the applied potential, hinting that the neutron-transparent PF_6^- is the main charge carrier in the system. A novel approach to isolate the concentration of species in solution with neutron imaging is discussed in the energy selective imaging section.

Main manuscript page 18:

When a positive potential is applied to the WE, the reverse reactions take place, and the resulting radiographs and the concentration profiles (Figure 5e) are mirrored compared to negative applied potentials. Because of the reactor architecture used in the NEUTRA experiments (stacked paper electrodes and a flow-by flow field) in combination with the low ionic conductivity of the electrolyte, there is only a small change in the cell capacity at negative potentials, consuming only ~8% of the total capacity (Figure S6), resulting in relatively low current densities. Therefore, at positive potentials, only a small amount of TEMPO is present in the WE compartment to be converted back to TEMPO^+ . As a result, large overpotentials are generated throughout the cell due to the low concentration of reactants to sustain the current. This explains the asymmetry in the current magnitudes when the polarity of the cell is reversed (i.e., +7 mA vs. -65 mA at 15.1 mL min^{-1} and $\pm 0.3 \text{ V}$, Figure 5b), and the even lower capacity recovery (~1-2%) resulting in underutilized capacity over the duration of the experiment (Figure S6).

When comparing the experiments with counter-ion BF_4^- and supporting salt KPF_6 , we can correlate the macroscopic performance with the concentration distributions through the reactor. For the KPF_6 experiments, all active species (i.e., TEMPO, TEMPO^+ , PF_6^-) are present in both compartments. Therefore, the macroscopic performance, i.e., the current output, is symmetric when operating at negative and positive applied potentials (Figure 4b) as the to-be-reacted species are present without the requirement of species crossover under the evaluated conditions (as the capacity change is limited to ~20%). The symmetric current output results in concentration profiles returning to the OCV profile when positive potentials are applied. Whereas for the BF_4^- experiment, the charged species are only present in one compartment (WE) initially and are required to cross the membrane to support the reactions, which is limited by the anion exchange membrane, resulting in asymmetric current magnitudes upon changing cell polarities (Figure 5b). The asymmetric current can be correlated to the concentration profiles as for positive applied potentials, the concentration does not fully return to the OCV profiles (Figure S7).

Main manuscript page 22:

These results corroborate the observations from the experiments in NEUTRA as only minor concentration fluctuations were observed with PF_6^- as supporting salt when an electric field is applied, whereas stark changes are detected when the supporting ion was changed to BF_4^- .

Main manuscript page 22:

*From the capacity curves (**Figure S9**), we observe that after the first potential step (-0.6 V at 21.1 mL min^{-1}), 60% of the total capacity is consumed. In the next step, after an applied potential of +0.6 V, only 45% of the total capacity is recovered (due to the lower current at set time), resulting in 15% underutilized capacity after a full polarization cycle because of the starting tank solutions (no BF_4^- in the CE) as explained in the transport of the counter-ion section. At the lower flow rate (6.7 mL min^{-1}), the capacity consumed at negative applied potentials is almost fully recovered at positive applied potentials, resulting in near symmetric current magnitudes. Interestingly, we find comparatively higher currents and capacity utilization with this reactor configuration in comparison with the reactor architecture used in the NEUTRA beamline (**Figure 5**), which can be correlated to the significant BF_4^- concentration fluctuations. We attribute these differences to the use of a different electrode material (a thick felt vs. a stack of thin carbon papers) and lower compressive forces. The higher porosity, apparent permeability and internal surface area of the felt electrode can explain the higher current densities observed in this reactor configuration.*

Furthermore, neutron radiography is a powerful method to obtain performance indicators of the reactor. Several were already mentioned in **Comment #1.1**, including the visualization of secondary phenomena in the reactor such as salt precipitation and incomplete filling of flow field channels, electrolyte infiltration in the porous electrode, or metal plating throughout the porous electrode. Other performance indicators that are possible to visualize using this approach are hydrogen evolution or the formation of local hotspots in the electrode. The visualization of such phenomena is powerful for a range of RFB types from the cell to stack level, for example zinc bromine plating at low current densities (as a result of the fast kinetics) for zinc-bromine batteries, vanadium precipitation in all-vanadium batteries, bromine evaporation in bromine-containing batteries, hydrogen evolution in all-iron batteries, and precipitation and cell filling in RFB stacks in general.

Finally, additional macroscopic properties could be extracted from neutron radiography experiments as was suggested by **Reviewer 1** in **Comment #1.2**, such as transport properties when clever experiments are designed or when coupling to computational models. To improve the quality of the manuscript and its impact on the RFB community we added a section concerning the practical relevance of the presented neutron radiography approach (page 24):

Practical application of the neutron imaging method

In this work, we have applied the new neutron imaging methodology to study non-aqueous redox flow batteries using multicomponent electrolytes. Beyond this specific application, we anticipate numerous applications of neutron imaging for quantifying concentration distributions in electrochemical cells and beyond. First, the resulting concentration maps can be used as experimental data to validate computational models that describe reactive mass transport. Here we used an electrolyte composed of a solvent, supporting electrolyte, and two redox active species - hence a complex, multicomponent system close to practical devices; but model experiments can be performed (e.g. with one or two analytes) to systematically deconvolute mass transport modes (e.g. diffusion, convection and migration) and their associated transport rates. Deeper fundamental understanding of reactive mass transport in

electrochemical reactors and through membranes will assist designing advanced electrochemical cells. Second, the methodology enables identification of local maldistributions in concentration, which can assist is designing better flow field geometries and electrodes, as well membrane crossover, Donnan exclusion and salt precipitation within the electrochemical cell, which are deleterious to performance and lifetime in several flow battery chemistries (e.g. non-aqueous, all-vanadium and all-iron). Third, we anticipate that the method - and adaptations on the detection physics - will be instrumental in advancing hybrid redox flow batteries (e.g. all-iron, zinc-bromine), where there are phase change reactions (e.g. plating and stripping or hydrogen evolution) fundamentally limiting the performance of the system. Fourth, we anticipate that the technique can be applied to technical systems such as electrochemical stacks, where traditional neutron imaging was instrumental in advance fuel cell stacks through visualization of water distributions. Fifth, beyond redox flow batteries, the method can be applied to other (electro)chemical reactors where concentration profiles determines performance such as electrochemical separations, flow chemistry, and chemical reactor design.

Finally, the use of molecularly engineered redox molecules acting as imaging probes might enable simultaneous visualization of concentration of multiple components (>2) when combined with energy-selective neutron imaging. Although the technique is still in its early stages, it displays considerable potential. We anticipate that ongoing advancements in neutron detectors and choppers will enable more sophisticated analyses of complex multicomponent systems using ToF-NI, offering enhanced spatial, temporal and energy resolution.

(Comment #2.2): The references need to be updated.

Author response: We thank the reviewer for pointing out the issue with the references and updated the references accordingly.

Reviewer #3: The authors developed a novel neutron radiography approach to quantify the concentration of species in solution during electrochemical operation in redox flow cells. They demonstrated for the first time the use of neutron radiography to image concentrations of redox active species and supporting salts in operando electrochemical flow cells. This is an excellent contribution to the field and the reported method and results are important guides for future development of advanced characterization for flow batteries. The schematics are excellent and very clear. I recommend to publish this with minor revision and here are my comments.

Author response: We thank the reviewer for their positive and insightful feedback and suggestions, which has improved the quality of the revised manuscript and the Supplementary Information. Please see below the detailed answers to each comment.

(Comment #3.1): It was mentioned “Using a low attenuating supporting salt (KPF₆), we selectively image the motion of redox active species...” this means that salt/active materials combination is important to ensure high selectivity in imaging? it would be helpful to discuss what are the materials constrains for salt, and active materials to be examined using this method, so readers can understand how to apply this method in their system.

Author response: We appreciate the reviewer for pointing out the need for clarification in our manuscript. The reviewer is correct - the combination of salt and active materials is important to ensuring high selectivity; however, it is nuanced since this depends on the specific imaging configuration. In our study, we employed two types of neutron imaging configurations: (1) white beam, thermal neutrons (NEUTRA beamline), and (2) cold neutrons coupled with time-of-flight (ICON beamline). The energy dependent neutron attenuation is easier to achieve for a colder neutron spectrum than a thermal one, due to the stronger interactions of cold neutrons with wavelengths comparable to the size of an atomic nucleus. Therefore, **ICON** utilizes cold neutrons and offers energy selective imaging by leveraging the energy-dependency of neutron attenuation in materials. This means that if the supporting salt and redox molecules have varying attenuations at different neutron energies, their concentrations can be distinguished, allowing for high imaging selectivity. On the other hand, **NEUTRA** relies on white beam imaging (fast neutrons), resulting in a neutron image that reflects the total neutron attenuation of all mixed components in the beam path.

Moreover, non-aqueous RFBs typically use fluorinated anions such as hexafluorophosphate (PF₆⁻) or tetrafluoroborate (BF₄⁻). **KPF₆**, composed of low attenuation elements for thermal neutrons, shows negligible neutron attenuation compared to hydrogenous redox molecules (TEMPO/TEMPO⁺). The transparency of KPF₆ enables selective imaging of redox molecules even in systems with complex reactive mass transport phenomena. However, if **BF₄⁻** is the supporting salt under white beam conditions, selective imaging of TEMPO/TEMPO⁺ becomes challenging due to similar attenuation coefficients between molecules, resulting in the neutron image representing the sum of BF₄⁻ + TEMPO + TEMPO⁺ concentrations.

To improve the quality of the manuscript and its impact on the RFB community the following additions have been made to the main manuscript:

Main manuscript page 13:

White beam neutron imaging does technically not allow selectivity towards a target component, but it is possible to obtain insights in concentration distributions of individual electrolyte species by careful selection of the redox active species and supporting salt, coupled with subtractive imaging.

Main manuscript page 13:

The electrolyte compositions used in this work provide selectivity towards both the redox molecules and supporting salt by utilizing salts with low and high attenuation with the neutron beam. On the other hand, minimizing the neutron attenuation of the redox active molecules through a reduced hydrogen content or deuterium labelling is another strategy to obtain contrast in multicomponent electrolytes. Combining both approaches would therefore be a powerful approach for future work to tune the neutron white beam imaging selectivity by transforming a complex multicomponent system into a single component system.

Main manuscript page 14:

Using this cell architecture and due to the negligible neutron attenuation of KPF_6 , we can track the movement of TEMPO between the electrodes.

Main manuscript page 17:

This resonates with the subtle changes in the concentration profiles in **Figure 4** as a function of the applied potential, hinting that the neutron-transparent PF_6^- is the main charge carrier in the system.

Supporting information page 8-9 Section S4:

Electrolyte composition

The composition of the salt and active materials, as well as their combination, is important to ensure high selectivity with neutron radiography. In this work, we studied non-aqueous RFBs that typically use fluorinated anions such as hexafluorophosphate (PF_6^-) or tetrafluoroborate (BF_4^-). KPF_6 is composed of elements having high neutron transmission and therefore shows negligible neutron attenuation compared to the redox active molecules employed in this study (TEMPO/TEMPO⁺). Hence, the low attenuation of KPF_6 enables the selective imaging of redox active molecules, even in systems with complex reactive mass transport phenomena. On the contrary, BF_4^- contains boron, which strongly attenuates the neutron beam. Therefore, when utilizing BF_4^- as the supporting salt under white beam conditions, selective imaging of TEMPO/TEMPO⁺ becomes challenging, resulting in the neutron image representing the sum of $BF_4^- + TEMPO + TEMPO^+$ concentrations. Accordingly, to guide the redox flow battery community in the right combinations of redox system and supporting salt, we listed the neutron attenuation with commonly used supporting salts and redox active molecules in **Table S2**.

Table S2: Neutron attenuation coefficients of commonly used supporting salts and redox active molecules used in redox flow batteries to show their potential for neutron radiography. The effective neutron attenuation coefficients were calculated for both the NEUTRA and ICON beamlines using a concentration of 1 M in deuterated acetonitrile (corrected for the attenuation of deuterated acetonitrile assuming the molecules are fully dissolved, and the volume of the solution does not change).

Common redox molecules	Chemical formula	Effective neutron attenuation coefficient NEUTRA [cm^{-1}]	Effective neutron attenuation coefficient ICON [cm^{-1}]
Anthraquinone	$C_{14}H_8O_2$	0.292	0.362
Phenazine	$C_{12}H_8N_2$	0.295	0.368
Cyclopropenium	$C_3H_3N_3$	0.600	0.757
Methyl phenothiazine	$C_{13}H_{11}NS$	0.384	0.481
Benzothiadiazoles	$C_6H_4N_2S$	0.1558	0.193
Methyl viologen	$C_{12}H_{14}N_2$	0.479	0.602
Ferrocene	$C_{10}H_{10}Fe$	0.344	0.432
Ferricyanide	$[Fe(CN)_6]^{3-}$	0.069	0.078
Chromium	$Cr, CrO_4^-, HCrO_4^-$	0.004, 0.013, 0.044	0.005, 0.015, 0.054
Iron	Fe	0.008	0.009
Hydrogen	H^+, H_2	0.031, 0.061	0.039, 0.078
Vanadium	V, VO^+, VO_2^+	0.006, 0.008, 0.010	0.008, 0.010, 0.013

Zinc	Zn^{2+}	0.003	0.004
Bromide	Br^-	0.006	0.009
Iodide	I^-	0.006	0.008
Polysulfide	S_x^{2-}	0.001	0.001
Heteropolyacid	$P_2W_{18}O_{62}^{6-}$	0.386	0.516
Riboflavin phosphate	$C_{17}H_{20}N_4O_9P^-$	0.716	0.894
Common supporting salts	Chemical formula	Effective neutron attenuation coefficient NEUTRA [cm^{-1}]	Effective neutron attenuation coefficient ICON [cm^{-1}]
KPF_6	KPF_6	0.019	0.020
$TBAPF_6$	$C_{16}H_{36}NPF_6$	1.174	1.483
$NaBF_4$	$NaBF_4$	0.429	0.714
$TBABF_4$	$C_{16}H_{36}NBF_4$	1.585	2.178
$LiTFSI$	$LiC_2NO_4F_6S_2$	0.077	0.106
$LiPF_6$	$LiPF_6$	0.055	0.082
$LiBF_4$	$LiBF_4$	0.466	0.776
Li^+	Li^+	0.039	0.065
TBA^+	$C_{16}H_{36}N^+$	1.158	1.466
$TFSI^-$	$C_2NO_4F_6S_2^-$	0.038	0.041
K^+	K^+	0.002	0.003
Na^+	Na^+	0.002	0.003
BF_4^-	BF_4^-	0.427	0.712
PF_6^-	PF_6^-	0.016	0.017
ClO_4^-	ClO_4^-	0.037	0.050
Cl^-	Cl^-	0.028	0.040
SO_4^{2-}	SO_4^{2-}	0.100	0.106
Hydroxide	OH^-	0.033	0.042

Moreover, not only the combination and composition of the salt and active materials is important, but also which specific imaging configuration is utilized. In our study, we employed two types of neutron imaging configurations: (1) white beam with thermal neutrons (NEUTRA beamline), and (2) cold neutrons coupled with the time-of-flight technique (ICON beamline). ICON offers neutron energy selective imaging by leveraging the energy-dependency of neutron attenuation in materials. This means that if the supporting salt and redox molecules have varying attenuations at different neutron energies, their concentrations can be distinguished, allowing for high imaging selectivity. On the other hand, NEUTRA relies on white beam imaging, resulting in a neutron image that reflects the total neutron attenuation of all mixed components in the beam path. Consequently, the electrolyte composition should depend on the imaging configuration of interest.

(Comment #3.2): *I am wondering would the materials selection for gasket/current collector affect the measurement?*

Author response: We thank the reviewer for this comment as it relates to critical aspects of the experimental design for successful neutron imaging of redox flow battery during operation. The importance of the battery housing material lies in the fraction of the neutron beam it will attenuate, the higher the attenuation of the neutron beam the lower the intensity of the collected neutron, the worse the image quality and imaging sensitivity. In our experimental setup, the neutron beam went through the gaskets made from PTFE and through the flow field material composed of graphite and polymer resin. Both materials are commonly used in redox flow batteries due to their physical and chemical properties. Where most of the elements in the periodic table have low neutron attenuation (**Figure R6**), few elements stand out due to their

high attenuation such as hydrogen, lithium, boron, cadmium and gadolinium [8]. From a chemical point of view, the use of PTFE as gasket materials reduces the beam attenuation compared to standard hydrocarbon polymers by replacing the highly attenuating hydrogen centres by fluorine atoms. On the other hand, flow fields are usually manufactured by compressing graphite and injecting phenolic resin in the remaining void space (~20% volume) [9]. Unlike for the gasket materials, the hydrogen atoms contained in the phenolic resin of the graphite flow fields do attenuate the neutron beam. To reduce the attenuation of the beam from the casing materials and to increase the sensitivity of the technique towards the sample, the flow field material distance in the beam path was reduced by creating some grooves in both the gaskets and flow fields (**Figure 1c-d**). To sum-up, the material selection for the casing materials should be carefully considered and the physicochemical requirements for the electrochemical application should overlap with the needs for proper neutron imaging.

Figure R6 – Neutron attenuation coefficients as a function of the atomic number for the elements of the periodic table. Adapted from Peng *et al.* [8].

Additionally, to guide the RFB community in choosing the right housing materials we included a table in the Supporting Information reporting the neutron attenuation of common materials used in RFBs as gaskets, current collectors etc. For the reviewer’s information, we additionally plotted the neutron attenuation and transmission values of the materials over the wavelength of the neutron beam in **Figure R7**.

Figure R7 – Neutron attenuation coefficient and transmission values plotted over the wavelength of the neutron beam for various commonly used materials in redox flow batteries.

Supporting information page 8 Section S4:

Section S4 – Experimental design considerations

Reactor materials and housing

When designing an experiment utilizing neutron radiography, the design of the electrochemical reactor needs to be considered. The importance of the material for the reactor lies in the fraction of the neutron beam it will attenuate, the higher the attenuation of the neutron beam, the lower the intensity of the collected neutron, and the lower the image quality and imaging sensitivity. In our experimental setup, the neutron beam went through gaskets made from polytetrafluorethylene and through flow fields composed of 3.18 mm thick resin-impregnated graphite plates, both commonly used in redox flow batteries due to their physical and chemical properties. Where most of the elements in the periodic table have a low neutron attenuation, few elements have a high attenuation with neutrons such as hydrogen, lithium, boron, cadmium and gadolinium. From a chemical point of view, the use of polytetrafluorethylene as gasket materials reduces the beam attenuation compared to standard hydrocarbon polymers by replacing the highly attenuating hydrogen centres by fluorine atoms. On the other hand, flow fields are usually manufactured by compressing graphite and injecting phenolic resin in the remaining void space (~20% volume). The hydrogen atoms present in the phenolic resin however do attenuate the neutron beam. To reduce the attenuation of the beam from reactor materials and housing and to increase the sensitivity of the technique towards the area of interest, the amount of reactor component material (flow field and gasket) in the direction of the neutron beam before the sample was reduced (**Figure 1c-d**). To sum-up, the material selection for the casing materials should be carefully considered and the physicochemical requirements for the electrochemical application should overlap with the needs for proper neutron imaging. The attenuation of the neutron beam for common reactor components is provided in **Table S1**.

Table S1: Neutron attenuation coefficients of commonly used reactor components to guide reactor design for neutron radiography experiments. The neutron attenuation coefficients were calculated for both the NEUTRA and ICON beamlines.

Reactor material	Chemical formula	Neutron attenuation coefficient NEUTRA [cm ⁻¹]	Neutron attenuation coefficient ICON [cm ⁻¹]
Polytetrafluorethylene gaskets	(C ₂ F ₄) _x	0.300	0.318
Polypropylene body	(C ₃ H ₆) _x	4.129	5.225
Graphite resin flow field (20% phenolic resin/ 80% graphite)	20% (C ₆ H ₆ O CH ₂ O) _x 80% C	1.058	1.270
Stainless steel	73% Fe, 18% Cr, 8% Ni, 1% Si	1.049	1.203
Aluminum	99% Al, 1% Mn	0.104	0.118

In addition, the following changes were made in the main manuscript:

Main manuscript page 13:

White beam neutron imaging does technically not allow selectivity towards a target component, but it is possible to obtain insights in concentration distributions of individual electrolyte species by careful selection of the redox active species and supporting salt, coupled with subtractive imaging.

Main manuscript page 13:

The electrolyte compositions used in this work provide selectivity towards both the redox molecules and supporting salt by utilizing salts with low and high attenuation with the neutron beam. On the other hand, minimizing the neutron attenuation of the redox active molecules through a reduced hydrogen content or deuterium labelling is another strategy to obtain contrast in multicomponent electrolytes. Combining both approaches would therefore be a powerful approach for future work to tune the neutron white beam imaging selectivity by transforming a complex multicomponent system into a single component system.

Main manuscript page 14:

Using this cell architecture and due to the negligible neutron attenuation of KPF₆, we can track the movement of TEMPO between the electrodes.

Main manuscript page 17:

This resonates with the subtle changes in the concentration profiles in **Figure 4** as a function of the applied potential, hinting that the neutron-transparent PF₆⁻ is the main charge carrier in the system.

(Comment #3.3): I am particularly interested in understanding if this technique can be applied in polysulfide based flow batteries (<https://doi.org/10.1038/s41560-023-01370-0>, <https://doi.org/10.1038/s41560-021-00804-x>) and heteropoly acid based flow batteries (<https://doi.org/10.1038/s41560-022-01011-y>), For polysulfide RFBs, for instance, it would be extremely interesting to examine the concentration profile near the electrode/membrane interfaces to understand the effect of membrane and molecular catalyst on the reaction. It would be helpful to elaborate in the perspective if possible.

Author response: We thank the reviewer for the interest in applying the method presented in this work to other RFB systems. As explained in **Comment #3.2**, the neutron attenuation of molecular materials is proportional to the sum of the elemental neutron attenuations. If a redox molecule only contains elements that lightly attenuate neutrons, tracking their concentration in the RFB will be challenging.

Table R1 shows the computed neutron attenuations of polysulfide, iodide, heteropoly acid, and riboflavin phosphate. As can be seen, applying neutron imaging to track polysulfide concentration is expected to be challenging due to the low attenuation of sulphur. Iodide on the other hand has slightly higher attenuation and could give contrast depending on the attenuation of the other electrolyte components and battery materials, whereas heteropoly acids are expected to give reasonable contrast due to their molecular structure and are therefore potentially compatible with neutron radiography. Finally, tracking the concentration of molecular catalysts for polysulfide RFBs (e.g., riboflavin phosphate) is in theory possible because of the high density of hydrogen atoms. However, the fact that the molecule would be present in catalytic amounts (<5 mM) would probably become challenging for neutron radiography. The attenuation coefficient and the transmission values versus the wavelength of the neutron beam is shown for the redox materials of interest for various concentrations in **Figure R8**.

Table R1 –Neutron attenuation coefficients of redox materials used in polysulfide-based flow batteries to show their potential for neutron radiography. The effective neutron attenuation coefficients were calculated for both the NEUTRA and ICON beamlines using a concentration of 1 M in deuterated acetonitrile (corrected for the attenuation of deuterated acetonitrile assuming the molecules are fully dissolved, and the volume of the solution does not change).

Redox material	Base structure	Effective neutron attenuation coefficient NEUTRA [cm ⁻¹]	Effective neutron attenuation coefficient ICON [cm ⁻¹]
Iodide	I _x ⁻	0.006	0.008
Polysulfide	S _x ²⁻	0.001	0.001
Heteropoly acid	P ₂ W ₁₈ O ₆₂ ⁶⁻	0.386	0.516
Riboflavin phosphate	C ₁₇ H ₂₀ N ₄ O ₉ P ⁻	0.716	0.894

Imaging concentrations in the vicinity of the membrane-electrode interface is challenging due to the low spatial resolution of neutron imaging coupled with interfacial scattering effects. Nevertheless, it is possible to observe concentration distributions a bit further away from the membrane and identify membrane effects on transport in the reactor area as explained in **Comment #1.1**. Additionally, constant advances in neutron imaging hardware and optical configurations help pushing the boundaries of what can be observed and extracted from neutron radiography experiments as highlighted in **Figure 7** of the main manuscript. By using a cold neutron spectrum coupled with the addition of a time-of-flight tube in the optical hardware, energy selective neutron imaging became possible. Finally, we envision that the ongoing research and development on neutron detectors with higher spatial resolution and choppers with faster duty cycles enables better neutron energy separation.

Additionally, to guide the RFB community in choosing the right combinations of redox system/supporting salt we included a table in the Supporting Information reporting the neutron attenuation of common supporting salt and redox molecules, see **Comment #3.1**.

Figure R8 – Neutron attenuation coefficient and transmission values plotted over the wavelength of the neutron beam for various redox materials over a range of concentrations, for: **(a)** iodide, **(b)** polysulfide, **(c)** heteropoly acid, and **(d)** riboflavin phosphate.

(Comment #3.4): Is the cell entirely sealed during the examination? As some organic molecules are sensitive to air/oxygen.

Author response: We thank the reviewer for the interest in the testing environment used during the battery testing and its possible influence on battery performance and neutron imaging. As rightfully pointed out by the reviewer, most organic redox molecules are air-sensitive making them impractical for oxygen-rich testing conditions [10,11]. During the battery testing the battery reactor was sealed using PTFE gaskets and the tank was sealed with a rubber septum with only openings for the tube inlet and outlets, thus limiting the oxygen availability in the system. In a previous work, we studied the stability of the electrolyte in similar conditions using a microelectrode [12]. In **Figure R7** we show the microelectrode analysis for a mixture of TEMPO/TEMPO⁺ over the course of 24 hours. We found that during this period, the microelectrode current varied by less than 2%, indicating good stability of the TEMPO/TEMPO⁺ under air-deprived (but not oxygen-free) conditions. This air stability is probably due to the high steric shielding of the TEMPO redox active sites coming from the methyl groups in α positions. In contrast, the presence of oxygen was found highly detrimental during our latest unpublished work for other organic molecules (e.g., 1,4-Di-tert-butyl-2,5-bis(2-methoxyethoxy)benzene, DBBB). To summarize, we are confident that in this particular case, the presence of oxygen in the system does not influence the electrochemical performance of the battery nor the neutron images. However, the studying of air-sensitive molecules with neutron imaging is feasible, for example by designing an air-tight system, purged with an inert gas.

Figure R7 – Microelectrode measurements from a 3-electrode sensor placed in the electrolyte tank of a TEMPO single cell electrolyte NAqRFB. Electrolyte mixture: 75% state of charge of TEMPO/TEMPO⁺ in 0.5 M TBAPF₆ in acetonitrile. Microelectrode current stability experiment over the course of 24 h with linear sweep voltammetry recorded every 5 min at 40 ml min⁻¹. The grey area represents one standard deviation for N=305. The positive currents correspond to electrochemical oxidation of TEMPO to TEMPO⁺, negative currents correspond to the reverse event. The graph was obtained from Jacquemond *et al.* [12].

To clarify and justify the experimental conditions, we made the following corrections in the manuscript:

Page 8, Section ‘Flow cell parts’:

Peristaltic pumps (Cole-Parmer) were used to pump the electrolyte to the cells with rubber tubes (Masterflex LS-14 tubing) connected to two separate 20 mL electrolyte tanks. The tanks were sealed with a rubber septum, limiting the oxygen availability in the system even though the presence of oxygen was not detrimental to the electrolytes employed [56].

(Comment #3.5): A minor point, the reference may have some issues showing “Error! Reference source not found.”.

Author response: We thank the reviewer for pointing out the issue with the references and updated the references accordingly.

References in the rebuttal

- [1] E.B. Boz, P. Boillat, A. Forner-Cuenca, *ACS Appl. Mater. Interfaces* 14 (2022) 41883–41895.
- [2] R.F. Savinell, N. Sinclair, X. Shen, J. Song, J.S. Wainright, in: *Flow Batteries*, John Wiley & Sons, Ltd, 2023, pp. 791–818.
- [3] K.L. Hawthorne, J.S. Wainright, R.F. Savinell, *Journal of Power Sources* 269 (2014) 216–224.
- [4] P. Kurzweil, M. Chwistek, *Journal of Power Sources* 176 (2008) 555–567.
- [5] M. van der Heijden, A. Forner-Cuenca, in: *Encyclopedia of Energy Storage*, Elsevier, 2022, pp. 480–499.
- [6] J.D. Milshtein, K.M. Tenny, J.L. Barton, J. Drake, R.M. Darling, F.R. Brushett, *J. Electrochem. Soc.* 164 (2017) E3265–E3275.
- [7] A.A. Wong, M.J. Aziz, S. Rubinstein, *ECS Trans.* 77 (2017) 153–161.
- [8] Z. Peng, Z. Tiejun, in: L.-O. Nilsson (Ed.), *Methods of Measuring Moisture in Building Materials and Structures*, Springer International Publishing, Cham, 2018, pp. 141–155.
- [9] GAB Neumann GmbH, *Heat Exchangers and Components in Graphite and Silicon Carbide*GAB Neumann GmbH (2017).
- [10] M. Pan, M. Shao, Z. Jin, *SmartMat* 4 (2023) e1198.
- [11] J.D. Milshtein, J.L. Barton, R.M. Darling, F.R. Brushett, *Journal of Power Sources* 327 (2016) 151–159.
- [12] R.R. Jacquemond, R. Geveling, A. Forner-Cuenca, K. Nijmeijer, *J. Electrochem. Soc.* 169 (2022) 080528.

REVIEWERS' COMMENTS

Reviewer #1 (Remarks to the Author):

The authors have successfully addressed my previous comments.

Reviewer #2 (Remarks to the Author):

The authors well addressed all the issues and it ca be accepted for publication.

Reviewer #3 (Remarks to the Author):

The authors have addressed my comments, I recommend to publish it.